# Assessment of Trace Metals Contamination, Species Distribution and Mobility in River Sediments Using EDTA Extraction

**DOI:** 10.3390/ijerph19126978

**Published:** 2022-06-07

**Authors:** Małgorzata Wojtkowska, Jan Bogacki

**Affiliations:** Faculty of Building Services, Hydro and Environmental Engineering, Warsaw University of Technology, Nowowiejska 20, 00-653 Warsaw, Poland; malgorzata.wojtkowska@pw.edu.pl

**Keywords:** EDTA leaching, bottom sediments, trace metals, heavy metals, speciation, residual trace metals

## Abstract

The impact of the ethylenediaminetetraacetic acid (EDTA) on speciation image of selected trace metals (Zn, Cd, Cu, Pb) in bottom sediments was determined. The influence on the effectiveness of metal removal of extraction multiplicity, type of metal, extraction time and concentration of EDTA were analyzed. With the increase of extraction multiplicity, the concentration of EDTA and contact time, the efficiency of trace metals leaching increased. The speciation analysis revealed that EDTA not only leached metals from bioavailable fractions, but also caused the transition of the metals between the fractions. The biggest amounts of bioavailable forms were found for Cd, less for Zn. The amount of bioavailable fraction was the lowest for Cu and Pb. The two first-order kinetic models fitted well the kinetics of metals extraction with EDTA, allowing the metals fractionation into “labile” (Q_1_), “moderately labile” (Q_2_) and “not extractable” fractions (Q_3_).

## 1. Introduction

The natural content of trace metals (TMs) in sediments is the result of the geochemical construction of the basin. Currently, the observed TM concentrations are typically much higher than the natural ones and are an effect of human activity, mainly the inflow of wastewater contaminated with TM and other industrial or agricultural activities. Industrial, agricultural and mining activities are important anthropogenic sources of TM pollution [1,2,3]. Firstly, deposited in soil, through surface runoff and groundwater, TMs reach rivers and lakes, where they are eventually deposited in bottom sediments. Accumulated in bottom sediments, metals pose a threat of secondary pollution of the reservoir. They do not undergo biological decomposition and are considered as persistent environmental pollution [4]. The migration and accumulation of metals and metalloids in the environmental components (sediment, soil) are largely dependent on pH, redox potential (Eh), types of minerals, organic carbon content and microorganisms, as well as plant species, root surface, exudate rate from roots and transpiration rate [5,6,7]. The immobilization and mobility of TMs in the soil environment or in bottom sediments strongly depends on their interaction with solid components, especially minerals, organic matter and microorganisms, which are the main components of solid phase aggregates [8,9]. Metal complexation with ligands and the adsorption to colloidal particles strongly influence their reactivity, mobility, bioavailability and toxicity [10]. During determination of the metal contamination degree, it is important to assess not only the total content of TM, but additionally, forms of their occurrence [11,12]. Metal ions in sediments are partitioned between the different phases, i.e., organic matter; oxyhydroxides of iron, aluminum and manganese; phyllosilicate minerals; carbonates and sulfides [13].

A speciation analysis is performed in order to determine the forms and types of compounds with which TMs are associated [14]. As a result of speciation analysis, the information on how much metal accumulated in the sediments is available for living organisms can be obtained [15]. There are many methodologies for speciation, because the methodology is designed according to the purpose to which it is aimed [16]. The two most common sequential extraction methods in use are the Tessier method and the BCR method [17,18,19,20,21,22,23]. In the case of high concentration of metals in the sediment, it is advisable to carry out reclamation works [24,25]. There are many methods of reclamation, one of which is the leaching of metals from sediments with a chelating agent [26,27,28], such as ethylene diamine tetraacetic acid (EDTA) [29,30,31,32,33,34,35,36,37,38]. The use of a mixture of chelating agents can improve the efficiency of the extraction [39].

The aim of this study was to determine the changes in the speciation image as an effect of EDTA concentration, extraction time and multiplicity for selected TMs. The originality and novelty of this manuscript lies in the comparison of three approaches that are generally rarely used simultaneously in research and described separately in the literature. 

## 2. Materials and Methods

### 2.1. Sampling Area

A sample was taken from the Utrata river, below the city of Pruszków, near the Góra Żbikowska landfill. The Utrata river is one of the most polluted rivers of the Mazovia Region (Mazovian Voivodeship in Poland) [12]. The source of pollution of the Utrata river has been industrial plants for many years discharging untreated wastewater directly into the river, as well as the unsecured landfill site. The river has its sources on the Northern slopes of the Rawska Highland. The river flows through, among others, Błonie, Pruszków and Żelazowa Wola. It is 76.5-km-long, it closes the catchment of 792 square km, its flow is 0.01–3.55 m^3^/s (in Pruszków—Góra Żbikowska landfill site, it is 0.3 m^3^/s), its mouth is located in Sochaczew and it flows into the Bzura river on the 26th km of its course. 

### 2.2. EDTA Extraction

The sediment was collected from the surface layer depth of 0–3 cm using a Kajak sampler (KC Denmark Research Equipment, Silkeborg, Denmark). A sample was transferred to a polyethylene container and then air-dried. The sample was then sieved through a set of sieves. Only the smallest fractions were collected for the tests. A sequential extraction EDTA was conducted on previously separated fraction with a diameter lower than 90 microns, due to the fact that TMs are accumulated predominantly in this fraction [40]. A total of 1 g of the air-dried Utrata river bottom sediments sample underwent EDTA sequential extraction. A 1 g sample of the sediment was placed in a 100 mL conical flask, and 20 mL of EDTA was added. The sample prepared was shaken on a laboratory shaker. Three different EDTA concentrations were tested: 0.01, 0.05 and 0.1 M. Nine extraction times were used: 1, 2, 3, 4, 6, 12, 24 and 48 h. After the specified time, the sample was filtered in order to separate the extract from the bottom sediment. For all EDTA concentrations and extraction times, single, double and triple extraction were performed, later described as extraction 1, extraction 2 and extraction 3. In the case of the double and triple extraction, the sample was air dried before the addition of the next EDTA. For double and triple extraction, each time, a fresh portion of 20 mL of EDTA was used. After triple extraction, the remaining TM content was also determined, later referred to as mineralization. All experiments were performed in triplicate. The result is given as the arithmetic mean of three measurements.

### 2.3. TM Speciation by Tessier Method 

Both in the bottom sediment prior to extraction, as well as in residual sediment after the 24 h extraction of TM, speciation was determined according to methodology of Tessier [41], as follows: the first stage for the speciation was extraction with 1 M MgCl_2_ in room temperature for one hour, with pH = 7 and speed of shaking of 250 rpm. After filtration and filtrate collection, the remaining residue was treated for 8 h in room temperature with 1 M CH_3_COONa acidified to pH = 5 with CH_3_COOH and a speed of shaking of 250 rpm. After filtrate collection, the third fraction was obtained by adding to the remaining residue, 20 mL 0.04 M NH_2_OH * HCl in 25% CH_3_COOH, at the temperature of 96 +/− 3 °C, for approximately 8 h. The fourth fraction was obtained by adding to the remaining residue of 3 mL 0.02 M HNO_3_ and 5 mL 8.8 M H_2_O_2_, at pH = 2, at the temperature of 85 +/− 2 °C, for 2 h. After this period the second portion of reagents was added (3 mL 8.8 M H_2_O_2_) and heated at the temperature of 85 +/− 2 °C, for 3 h. After cooling, 5 mL of 3.2 M CH_3_COONH_4_ was added in 20% HNO_3_ and stirred for 30 min, with a speed of shaking of 250 rpm. The bottom residue remaining after the filtration was mineralized with an HNO_3_/HClO_4_ mixture in a 5:1 ratio. As a result of the extraction, five fractions were obtained, referred to as exchangeable, carbonate, Fe/Mn bound, organic and residual. A detailed scheme of speciation is shown in Figure 1. The content of cadmium, lead, copper and zinc in every fraction was determined with Flame Atomic Absorption spectrometry (FAAS) using AAnalyst 300 (Perkin Elmer, Waltham, MA, USA).

All reagents used to prepare the extracting solutions were products of analytical-grade quality. Water of high purity obtained from a Millipore apparatus (water resistivity = 18 MΩ∙cm) was used for solution preparation. The flasks were cleaned in 1 mol/L nitric acid and then rinsed with pure water. A digestion block was used for mineralization of the soil samples and for the sequential extraction protocol.

### 2.4. Extraction Kinetics

According to Fangueiro et al. [42], Santos et al. [43] or Li et al. [44], multiple first-order extraction reactions may take place simultaneously, having rates that are assumed to be independent of each other. Each reaction rate can, thus, be expressed as (1):(dq_i_)/dt = k_i_ (Q_i_ − q_i_) (1)
where q_i_ represents the quantity of desorbed metal from binding location i, per mass unit of soil (mg/kg dry mass), after time t (h); Q_i_ represents the quantity of desorbed metal from i, per mass unit of soil (mg/kg dry mass), at equilibrium, and k_i_ is the rate constant of the extraction reaction for compartment i (1/h). 

The model of multiple first-order reactions allows determination of the quantity and the extraction rate of metal cations associated to each fraction. A simplification of the presented model to a two first-order reactions model enables classification of each metal cation into three fractions, Q_1_, Q_2_ and Q_3_ [43]. The use of the above equation allows the characterization of the three fractions, as follows [42] (2):Q = Q_1_ + Q_2_ + Q_3_
(2)

Q: pseudo-total concentration of metal in the sediment [mg/kg dry mass].

Q_1_: ‘‘labile” fraction [mg/kg dry mass], 

Q_2_: ‘‘moderately labile” fraction [mg/kg dry mass],

Q_3_: metal fraction which is not extractable [mg/kg dry mass].

For the purposes of the calculations, it was assumed that the first two fractions obtained during the extraction according to the Tessier method are “labile”, and the last one is “not extractable”. Extraction kinetics and statistical calculations were conducted using MS Excel. Statistical relationships were calculated by grouping the mass removal of selected TMs into pairs for all fractions, contact times, EDTA concentrations and extraction multiplicities.

## 3. Results

### 3.1. Raw Sediment Fraction Analysis

Particle size fractions were determined in air-dried bottom sediment samples. The content of various grain size fractions is shown in Table 1.

The analyzed samples consisted mainly of fine particles. The fraction with a grain size below 90 μm was 41% of the total sample. The coarse fraction, above 500 microns, was only about 15%. The sample was rich in plant elements, in the form of clearly visible leaves, stems and roots. The main component of the coarse fraction were fragments of plants.

### 3.2. TM Speciation in Raw Sediment

TM speciation determined in the raw sediment is shown in Table 2. Metals that are weakly bound could be available for living organisms and are bioavailable. Usually, the first three fractions (exchangeable, carbonate and Fe/Mn bound) are considered to be bioavailable.

Pb-dominant fractions were carbonate- and Fe/Mn-bound ones, and they were 27.0% and 27.8% of total content, respectively. In case of Cd, the highest content was determined for the exchangeable and carbonate fraction: 47.0% and 36.2%. The Cd amount in other fractions was relatively low. For Zn, the most important fraction was the carbonate one, with 38.9% of total mass. Cu was accumulated in the organic fraction in 71.3%. It is known that Cu is strongly bound by organic matter (humic substances) [43]. 

According to Polish legal regulation, sediments collected from the Utrata river have to be considered as Pb, Cu and Cd polluted [45], as it was shown in Table 3.

### 3.3. EDTA Sequential Extraction

#### 3.3.1. Lead

The results of Pb extraction with EDTA are shown on Figure 2a–f. The amount of TM removed in a single extraction is described as extraction 1, the amount of TM removed in a double extraction is described as extraction 2 and the amount of TM removed in a triple extraction is described as extraction 3. The TM content remaining after the triple extraction was marked as mineralization. Analogous notations were used to describe the figures representing the effectiveness of residual TM removal. With the increase of EDTA concentration, Pb leaching effectiveness increased significantly, ranging from 45% to 95% for 0.01 M to 0.1 M, respectively.

Increasing contact time had a low impact on the leaching effectiveness and was mainly observed for 0.01 M EDTA. The difference between 1 h and 48 h contact time was approximately 10%. At higher concentrations (0.05 and 0.1 M), there was no difference between 1 h and 48 h contact time.

What had the most significant impact on the Pb leaching effectiveness was the extraction multiplicity. This effect was greater for a lower EDTA concentration. For 0.01 M EDTA, each portion of the reagent allowed leaching of 10–20% of the total Pb content. In the case of the 0.05 M concentration, the first dose of the reagent allowed about 50% of Pb to be removed. This result, however, was reduced by half with each subsequent dose. The first dose of 0.1 M EDTA removed over 70–80% of the total Pb content. Each subsequent dose resulted in much lower removal; the second dosage was twice as effective as the third.

Use of 0.01 M EDTA, in the case of Pb, was inadequate to remove the metal from the solid. Even the use of three extractions did not result in decreasing Pb content below 100 mg/kg dry mass, which makes soil still contaminated with Pb. The situation was similar in the case of extraction with 0.05 M EDTA. However, in the case of extraction 2 and 3 and in the case of 0.1 M EDTA, the Pb content decreased to a level regarded as uncontaminated.

#### 3.3.2. Cadmium

The results of Cd extraction with EDTA are shown on the Figure 3a–f. With the increase of EDTA concentration, the effectiveness of Cd leaching increased significantly, ranging from 40% to 95% for 0.01 M to 0.1 M respectively.

Increasing contact time had a significant impact on the Cd leaching effectiveness at a lower EDTA concentration. Increasing EDTA concentration resulted in decreasing the contact time influence. The difference in leaching efficiency between 1 h and 48 h contact time was 30%, 15% and 10% for EDTA of 0.01, 0.05, 0.1 M, respectively. Extraction multiplicity has the most significant impact on the Cd leaching effectiveness. This effect was greater for a lower EDTA concentration. For 0.01 M EDTA, every time, a new portion of the reagent allowed leaching of 10–20% of the total Cd content. Especially in the case of the first extraction, the effect of extended contact time was visible. The effectiveness of the leaching was 15% for 1 h, but for 48 h, it was 45%. Each additional portion of reagents resulted in the removal of a smaller amount of Cd than previously. Moreover, the effect of increasing contact time on the Cd removal efficiency was negligible. For 0.05 M EDTA, the first portion of the reagent leached most of Cd, and the efficiency of the next dose was significantly worse than that of the previous one. The effectiveness of leaching with the first dose was clearly dependent on the time, with 50–75% for 1 h and 48 h, respectively. An impact of contact time prolongation for the next dose was not observed. In EDTA of 0.1 M concentration, the efficiency of the first dose was time-dependent and reached 60% for 1 h and 75% for 48 h. For extraction times longer than 4 h, no difference in leaching efficiency was observed. The efficacy of each next dose was clearly worse than the previous one, with 15% and 7% for the second and the third one, respectively.

In the case of Cd, regardless of the multiplicity and EDTA concentration, the metal content after extraction was decreased to below 4 mg/kg dry mass. The sediment could be regarded as not polluted with Cd. 

#### 3.3.3. Zinc

The results of Zn extraction with EDTA are shown in Figure 4a–f. With the increase of EDTA concentration, the effectiveness of Zn leaching slightly increased, ranging from 85% to 90% from 0.01 M to 0.1 M, respectively. Increasing contact time did not affect the efficiency of leaching. There was no difference in the effectiveness between 1 h and 48 h leaching time for all concentrations of EDTA. The only factor that had a significant influence on leaching efficiency was extraction multiplicity, and as it was observed also for Cd and Pb, this effect was greater for a lower EDTA concentration. The efficiency of the first dose was the highest, with leaching effectiveness decreasing with the dose number. The first dose reached, according to the EDTA concentration, 45–50%, the second reached about 20, and the third one reached approximately 15% of the total Zn contained in the sample. The differences in leaching effectiveness between the EDTA concentrations of 0.1 M and 0.01 M were very small.

Due to the initial low concentration of Zn, the sediment from the beginning was regarded as not polluted. Sequential extraction additionally allowed the Zn concentration in the sediment to decrease to 20–125 mg/kg dry mass.

#### 3.3.4. Copper

Results of Cu extraction with EDTA are shown in Figure 5a–f. With the increase of EDTA concentration, the effectiveness of Cu leaching increased significantly, ranging from 50 to 88% for 0.01 M and 0.1 M, respectively.

With increasing contact time, leaching efficiency also increased. The difference between 1 h and 48 h leaching time was 15%, 5% and 10% for 0.01 M, 0.05 M and 0.1 M, respectively. Extraction multiplicity has the highest impact on the leaching effectiveness. This effect was greater for a lower EDTA concentration. For 0.01 M EDTA in any subsequent leaching, a new portion of EDTA allowed for leaching of an additional 10–15% of the total Cu content. The effect of contact lengthening was especially visible in the case of the first extraction. Leaching effectiveness ranged from 30% for 1 h to 45% for 48 h. Each additional portion of the reagent resulted in the removal of a smaller amount of Cu. The impact of extended contact time on the effectiveness of Cu removal was low, and the differences in effectiveness were driven by the first dose effect.

In the case of the 0.05 M EDTA, the first portion of the reagent leached most of Cu, and the efficiency of each dose was significantly worse than the previous one. The effectiveness of washing the first dose of the reagents was clearly dependent on the time. For 1 h, it was 50%, and for 48 h it was 60%. An impact of contact time prolongation for the next dose was not observed.

For EDTA in a concentration of 0.1 M, the first dose efficiency was time-dependent and reached 60–75% for 1 and 48 h, respectively. The efficiency of each next dose was clearly worse than the previous one. In the case of the second, it was about 15%, and for the third, it was about 10%. For extraction times longer than 4 h, no differences in leaching efficiency for the third EDTA dose were observed.

The application of 0.01 M EDTA in extraction 1 was insufficient to reach Cu removal to below 150 mg/kg dry mass. For all other cases, a 150 mg/kg dry mass level was reached, and sediment should be regarded as unpolluted.

### 3.4. Lability Determination

Previous studies have shown [42,43] that 24 h extractions are sufficient to allow reaching of an equilibrium state in sediments [31]. Santos et al. [43] found that according to Labanowski et al. [15], the readily ‘‘labile” pool using EDTA overestimates the leachability of metals in soils, particularly in the case of Pb. They observed that Pb was the least mobile metal in soil, but it was the one with the highest ‘‘labile” pool, determined by EDTA extraction. According to the literature, EDTA releases Pb from several soil compartments, particularly Pb associated to Fe and Mn oxides and to organic matter. Santos et al. [43] found that there is a lack of studies about the relationship between kinetically labile fractions and EDTA extraction and fractions associated with different soil compartments, determined by sequential extraction. However, Gismera et al. [46] obtained a good correlation between the Zn labile fraction (Q_1_), associated to EDTA, and the exchangeable and carbonate-bound fraction determined by the BCR sequential extraction scheme. 

In order to simulate the kinetics of metal desorption, Fangueiro et al. [42] proposed a model describing three fractions: “labile”, “moderately labile” and “not extractable”. The contents of “labile”, “moderately labile” and “not extractable” fractions, based on kinetic models for different TM end extraction multiplicity, are shown in Table 4.

A general two-step desorption process was observed in the variability of the metal cation desorption rate in all samples, indicating two kinetically distinguishable pools, which correspond to two metal fractions characterized by two different desorption rates, i.e., a high desorption rate at the start of leaching followed by a reduction in reaction time.

The “labile” (Q_1_) and “moderately labile” (Q_2_) metal fractions were estimated by kinetic simulation for all samples (Table 4). The contents of the labile and less labile fractions are expressed as fractions of the sum of the sediment content for comparison with the metal fractions obtained by single speciation for different EDTA concentrations and being made three-fold.

For all samples tested, the total Pb-EDTA fraction corresponded to 15% to 71% of the labile pool; similarly, for Cd-EDTA, the fraction ranged from 15% to 75% of the labile form, while Zn-EDTA and Cu-EDTA corresponded to 51% and 37%, respectively. The indelible metal fraction, Q_3_, reached the median for Cd (37%), Cu (60%) and for Zn and Pb (49%).

The unstable and less labile fractions represent an operationally defined concentration of metal desorbed from various chemical components of the bottom sediments. Contrary to the Tessier diagram, this method does not include the actual sediment fraction, such as iron oxides or carbonates, but it can be used to predict the total labile fraction.

### 3.5. Extraction Correlations

The correlations between the extraction of metals from bottom sediments and the EDTA concentration for different metal-to-metal pairs are shown in Figure 6.

A significant correlation for extraction 1 between Cd and Cu of 0.99, 0.84 and 0.79 for decreasing EDTA concentrations, respectively, was observed. A similar pattern was observed for Cu and Pb. For Zn, no correlation with other metals was observed. Sun et al. [4] noted that during extraction, EDTA leached metals in proportion to their content in the sediment.

### 3.6. Speciation of Metals in Sediment after EDTA Extraction

In the conducted research, extraction was carried out for up to 48 h. Earlier studies [15] show that 24 h is a time sufficient to establish an equilibrium state; however, some authors suggest that this time is much longer, even 216 h [39]. Zhang et al. [35] drew attention to the possibility of changing the form of trace metals after extraction, but they used low concentrations of EDTA (0.005–0.0005 M) and short extraction times (0.5–2.0 h), which did not allow to accurately measure these changes. In the author’s own research, speciation of the sediments was performed for three repetitions of extraction with EDTA 0.01 M, 0.05 M and 0.1 M solutions.

#### 3.6.1. Lead

The speciation of Pb after EDTA extraction is shown in Figure 6a,b. As a result of EDTA 0.01 M, a leaching decline of bioavailable fractions was observed, especially for carbonate and Fe/Mn-bound ones. In the case of the organic fraction for extraction 1, the increase from 33 to 45 mg/kg dry mass was observed, while for extraction 2 and 3, the amount was decreased. In relation to the residual fraction, slight increases were observed, up to 28 and 35 mg/kg for extraction 1 and 3 respectively, in comparison to 27 mg/kg for raw sediment. The usage of 0.05 M EDTA resulted in decreasing concentrations of exchangeable, carbonate and Fe/Mn-bound fractions. In the case of the organic and residual ones, for extraction 1, increasing up to 38 and 42 mg/kg was observed, but for extraction 2 and 3, decreasing was observed. The usage of 0.1 M EDTA caused decreasing concentrations for all factions, with the largest for carbonate and Fe/Mn-bound and the smallest for the residual.

#### 3.6.2. Cadmium

The speciation of Cd after EDTA extraction is shown in Figure 6c,d. For 0.01 M EDTA, a clear decline for exchangeable and carbonate fraction was observed. In the case of Fe/Mn-bound, Cd content remained at the same level for extraction 1 and 2 or slightly increased. The content of the organic fraction increased and reached from 0.5 mg/kg for extraction 1 to 0.3 mg/kg for extraction 3. The residual fraction content decreased with respect to the raw sediment, and the decreasing rate increased with the multiplicity of extraction. For 0.05 M EDTA, the exchangeable and carbonate fraction content was significantly decreased. The amount of the Fe/Mn-bound and organic fraction after extraction 1 slightly increased, whereas for extraction 2 and 3, it decreased. The residual fraction decreased with the multiplicity of extraction. Extraction with 0.1 M EDTA resulted in decreasing the exchangeable, carbonate and Fe/Mn-bound fractions. The content of exchangeable and Fe/Mn-bound fractions were smallest after extraction 2. The amount of organic fraction for extraction 1 was higher than in raw sediment; for a bigger multiplicity, it was lower. The residual fraction decreased with the multiplicity of extraction. 

#### 3.6.3. Zinc

The speciation of Zn after EDTA extraction is shown in Figure 6e,f. In the case of 0.01 M EDTA extraction, Zn was leached proportionally to the extraction times of almost all fractions. For the exchangeable fraction after extraction 1 and 2, the Zn content in the treated sediment was higher than in the raw one. As a result of extraction 1, a small increase in the residual fraction was also observed. For 0.05 M EDTA extraction, as well as for 0.01 M, proportional Zn leaching with a multiplicity of extraction was observed. For extraction 1 and 2, a significant increase, three- and twofold, in the exchangeable fraction was measured. In the case of 0.1 M EDTA Zn leaching, proportional leaching to the multiplicity of extraction was observed. For the exchangeable fraction, a significant increase in the content in relation to raw sediment of 3.5- and 2.5-fold was observed. As a result of extraction 1, a twofold increase in the residual fraction in relation to the raw sediment was observed. In the case of Zn, the most important factor determining the leaching effectiveness was the multiplicity of extraction. The differences between the metal content in the individual fractions for the same times and different EDTA concentrations were very small. For all EDTA extractions, the concentration amount of exchangeable fraction in treated sediment was higher than in the raw one.

#### 3.6.4. Copper

The speciation of Cu after EDTA extraction is shown in Figure 7g,h. As a result of leaching with 0.01 M EDTA, the content of the exchangeable, carbonate and Fe/Mn-bound fraction was maintained at the same level, with a slight downward trend with increasing extraction multiplicity. The removal of the organic fraction increased with multiplicity extraction. In raw sediment, as well as in the treated one, Cu accumulated mostly in the organic fraction. Regardless of extraction multiplicity, a constant level of residual fraction, 1.5-fold bigger than in raw sediment, was observed in the treated sediment. The usage of 0.05 M EDTA allowed all fractions to decrease contents in a relationship with the extraction multiplicity. The removal of the exchangeable fraction was the lowest, and the content of this fraction, regardless of multiplicity, was higher than in the raw sediment. The content of the residual fraction was decreasing very slowly, and even multiple extractions removed only a small part of it. In the case of 0.1 M EDTA, the amount of the exchangeable, carbonate, organic and reducing fraction decreased with an increasing extraction multiplicity. Exchangeable fraction content in the treated sediment was much higher than the raw sediment, regardless of the extraction multiplicity. The content of the residual fraction in the treated sediment, regardless of the extraction multiplicity, was similar to its amount in the raw sediment; the smallest content was observed after extraction 2. For 0.1 M EDTA, dominance of the organic fraction, as a fraction gathering the biggest amount of copper in the sediments, ceased to be significant. For extraction 3, it was no longer a dominant fraction. Regardless of the EDTA concentration, a significant increase in the exchangeable fraction in the sediment was observed. EDTA at high concentrations poses a very low ability for removing Cu from the residual fraction, whereas at low concentrations, it even causes an increase.

## 4. Discussion

A bottom sediment sample was taken for the tests. This means that the material in its original state and environment was in constant contact with water. It can be suspected that all the compounds soluble in water dissolved in it, and the overwhelming majority of the sample components were compounds with limited water solubility. The sample, in accordance with the methodology, was dried and then extracted with aqueous solutions of various chemical compounds. During the extraction, due to the addition of the solvent, the chemical compounds constituting the sample could dissolve. However, as they are compounds with limited solubility, it was assumed that the effect of the solvent itself would be negligible. It seems that the influence of the chemicals used during extraction will be much more important here. It should be noted that EDTA is a weak acid and could cause direct chemical reactions with the components constituting the sample. This means that many compounds, and products of chemical reactions, had greater solubility than the parent compounds. On the one hand, the occurring chemical reactions will cause the release of metal ions into the solution; on the other hand, however, they will cause the structure of the sample to remodel, which will result in a change in the speciation image. Some of the chemical reactions taking place are quite slow, which means that the assumed extraction times are too short to complete the reaction completely. This hypothesis was confirmed by the effects of the second and third extraction. After each subsequent extraction, the speciation pattern continues to change, and the mass of the sample remaining after the process should decrease. Nevertheless, this is only speculation, since the sample was not weighed after the process.

EDTA is a very strong metal chelating agent. Increasing its concentration, contact time and the extraction multiplication factor significantly contributes to increasing the efficiency of TM removal from bottom sediments. This relationship was observed for all tested metals. The effectiveness of the extraction process depended on the properties of the metal and its speciation in the sediment.

EDTA extraction allows not only the fractions recognized as bioavailable (exchangeable, carbonate, Fe/Mn-bound) to be removed, but also biologically not available forms (organic and residual). The research revealed a different influence of the above-mentioned factors on the effects of the extraction process for the four metals were tested. The concentration of EDTA had the strongest effect on lead leaching. Only in the case of the lowest concentration, the extension of the extraction time and the multiplicity of the extraction had an impact on the efficiency of Pb elution. Cd extraction increased with time for each EDTA concentration, but it was the strongest with the 0.1 M EDTA solution. The Cd extraction fold was also the most significant for the 0.1 M EDTA solution. The Zn extraction was the same for the three EDTA concentrations and did not change significantly with time. As for Cd, the extraction of Zn from the sludge was most influenced by the extraction multiplication factor. Cu extraction increased with an increasing EDTA concentration and during extraction. The copper leaching rate was the same for all three EDTA solutions. The obtained results suggest that EDTA may be an appropriate factor to determine the content of bioavailable TM in sediments. The ability to extract Zn and Cd by EDTA compared to Pb and Cu may be affected by anoxia, which results from the greater solubility of ZnS and CdS compared to CuS and PbS [47].

The use of EDTA also changes the image of the speciation of the remaining sediment. The expected effect was the leaching of metals from weakly bound fractions. However, a shift of metals from the stronger metal-binding fraction to the weaker metal-binding fraction was also observed. This phenomenon should be associated with a change in the composition and content of substances contained in the bottom sediment remaining after extraction. EDTA has the ability to remove not only metals but also other substances of sediments. The change in the matrix composition, therefore, translated into a change in the speciation image of metals. As a result, it is advisable to use another EDTA extraction cycle. In the next extraction cycle, metals in weakly bound fractions are removed, and the remaining sediments are further transformed, which results in the shifting of TM from the fractions that bind the metals more strongly to the fractions that bind them weaker. However, it should be remembered that each subsequent extraction cycle rinses not only TM from the sediments, but also other substances, including carbonates, phosphates, organic compounds, etc. [48]. After extraction, some TMs remain in the sediment, and they are still bioavailable. The highest amounts of bioavailable forms were found for Cd and less for Zn. The amount of the bioavailable fraction was the lowest for Cu and Pb. Although the use of EDTA reduces the TM content in the treated sludge, due to a change in the matrix composition, it may lead to an increase in the toxicity and bioavailability of metals remaining afterwards [48]. Although EDTA washing could effectively remove TM, it also may result in a significant decline in sediment quality [49]. The obtained results are consistent with the ones obtained by other researchers. Cheng et al. [50], by using a blend of 0.05 M EDTA with weak organic acids (citric, oxalic and tartaric), were able to remove more than 80% of TMs. Simultaneously, using the BCR extraction scheme, Cheng et al. [50] confirmed a change of the speciation image after EDTA extraction. The removal of TMs mainly from the labile fractions was observed, but also, a significant reduction in the content of bound metals in the last, fourth and residual fraction was observed. They suggested that acidification of the environment increases the efficiency of TM extraction. Ferrans et al. [51] came to similar conclusions using EDTA to remove metals from dredged sediments. Using 0.05 M EDTA, they removed more metals than they did with 0.01 M. Lumia et al. [52] reported up to 85% removal of TM with the use of EDTA. The extraction efficiency was metal-dependent, and it was the highest for Zn. The effectiveness of 0.1 M EDTA was greater than using the 0.05 M solution. However, increasing the EDTA concentration to 1 M did not increase the leaching efficiency, as opposed to increasing the contact time from 3 to 24 h. The discussed results show the influence of bottom sediment components, such as minerals and organic substances, on the adsorption, immobilization and availability of TMs, which was confirmed by other researchers [53,54,55,56]. The extraction processes remove organic substances present in the sediments. Under these conditions, non-eluted TMs bind in the crystal structures of the bottom sediment [8]. Metals bound in the mineral structures are also mobile and may migrate to other components of the environment [57]. 

## 5. Conclusions

The originality and novelty of this manuscript lies in the comparison of three approaches that are generally rarely used simultaneously in research and are described separately in the literature. The metals were characterized by abundant extraction in more concentrated EDTA solutions and in the first stages of extraction. This indicates a high constant of the desorption rate and a high extraction by EDTA.

The method of sequential extraction of metals from environmental samples provides important information about the possible mobility and toxicity of metals to environmental components. The speciation analysis performed in the remains after EDTA extraction showed the migration of TM between the sediment fractions. This was especially observed in the case of copper, which has a very strong affinity for the organic fraction, and the tests performed in the remaining extraction sediments increased its share in the residual fraction. Such behavior indicates a strong dependence of the share of metals in individual fractions on the mineral–organic–microbiological composition of bottom sediments.

The extraction multiplicity and EDTA concentration, as well as the contact time, affect the results of TM leaching from the bottom sediments. For the same contact time and the same EDTA concentration, the highest leaching efficiency was obtained for Cd, lower for Pb and Cu and the lowest for Zn.

The kinetic approach revealed the presence of two distinct pools in the metal-EDTA fractions and a higher desorption rate for Cd and Zn compared to Pb and Cu. Two first-order kinetic models fitted well with EDTA metal extraction kinetics, allowing metals to be fractionated into “labile” (Q_1_), “moderately labile” (Q_2_) and “not extractable” fractions (Q_3_). The correlation study showed some significant metal synergy, which was quite varied in the sediment fractions.

Three methods were used in this work: the sequential extraction scheme according to Tessier and an EDTA extraction and kinetic calculations, which showed clear differentiation in the availability of four tested TMs occurring under the same conditions in the tested sludge.

## Figures and Tables

**Figure 1 ijerph-19-06978-f001:**
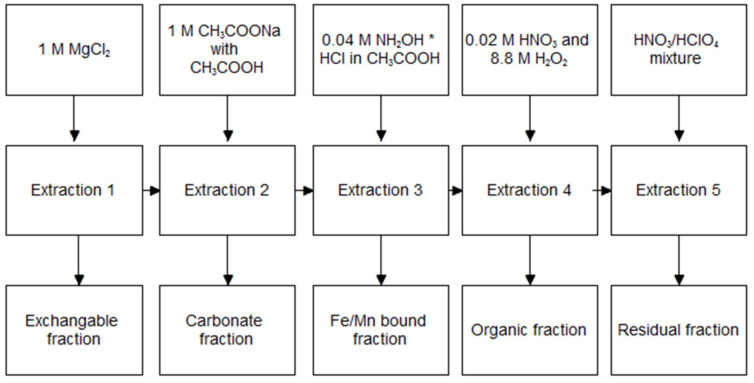
Tessier speciation scheme.

**Figure 2 ijerph-19-06978-f002:**
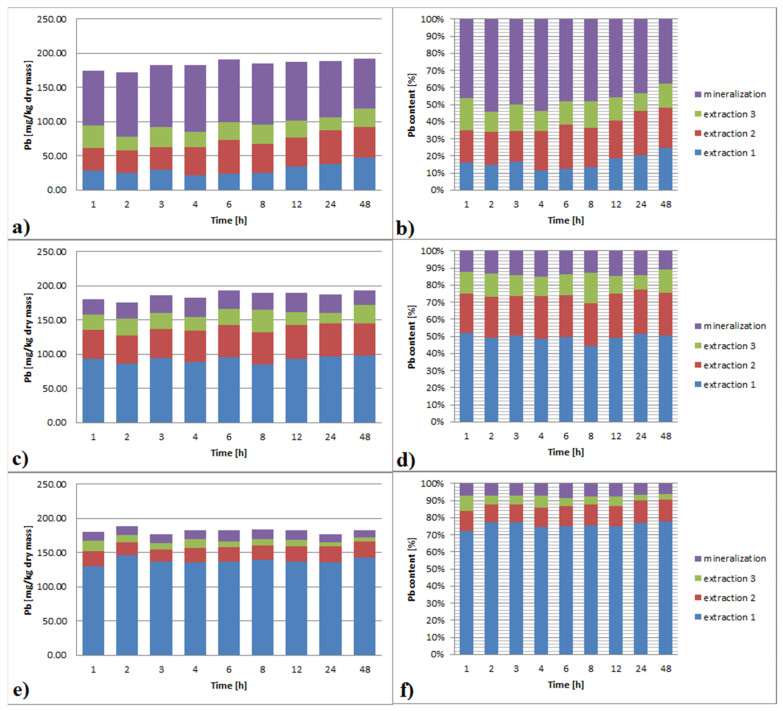
(**a**–**f**). Pb extraction from bottom sediments with different EDTA concentrations (**a**,**b**): 0.01 M; (**c**,**d**): 0.05 M; (**e**,**f**): 0.1 M and extraction multiplicity.

**Figure 3 ijerph-19-06978-f003:**
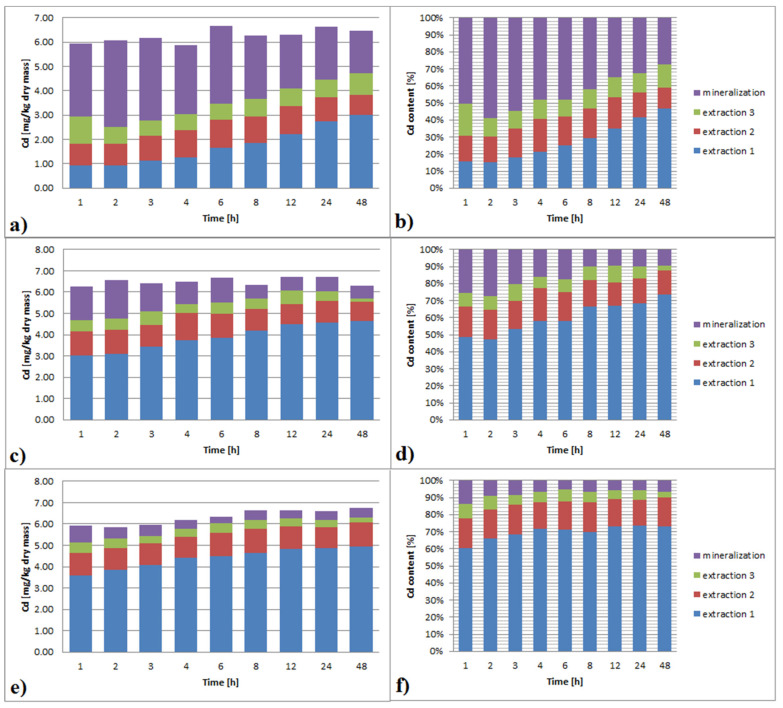
(**a**–**f**)**.** Cd extraction from bottom sediments with different EDTA concentrations (**a**,**b**): 0.01 M; (**c**,**d**): 0.05 M; (**e**,**f**): 0.1 M and extraction multiplicity.

**Figure 4 ijerph-19-06978-f004:**
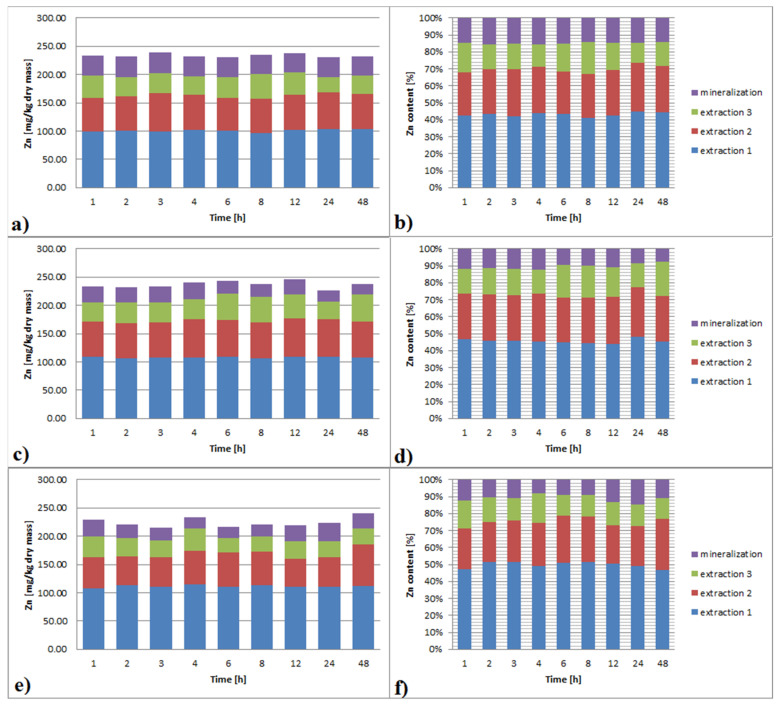
(**a**–**f**). Zn extraction from bottom sediments with different EDTA concentrations (**a**,**b**): 0.01 M; (**c**,**d**): 0.05 M; (**e**,**f**): 0.1 M and extraction multiplicity.

**Figure 5 ijerph-19-06978-f005:**
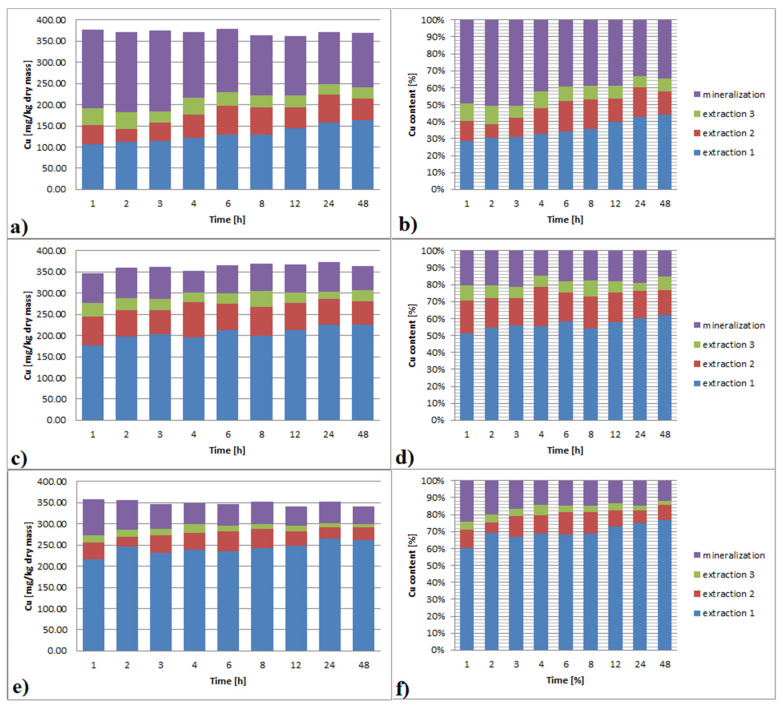
(**a**–**f**)**.** Cu extraction from bottom sediments with different EDTA concentrations (**a**,**b**): 0.01 M; (**c**,**d**): 0.05 M; (**e**,**f**): 0.1 M and extraction multiplicity.

**Figure 6 ijerph-19-06978-f006:**
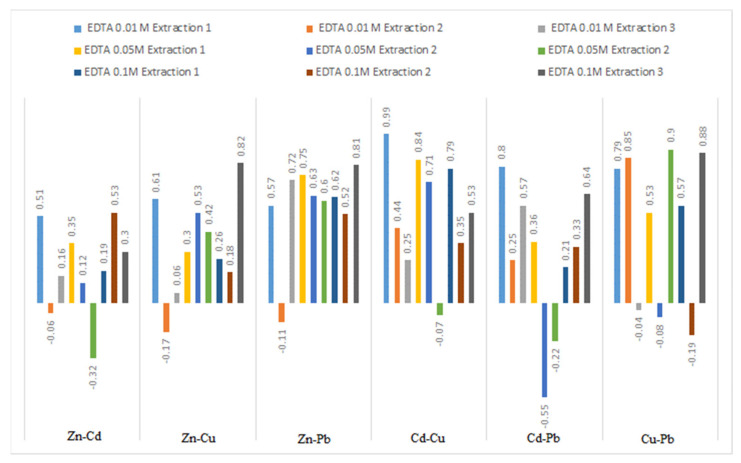
Correlations between trace metals extraction.

**Figure 7 ijerph-19-06978-f007:**
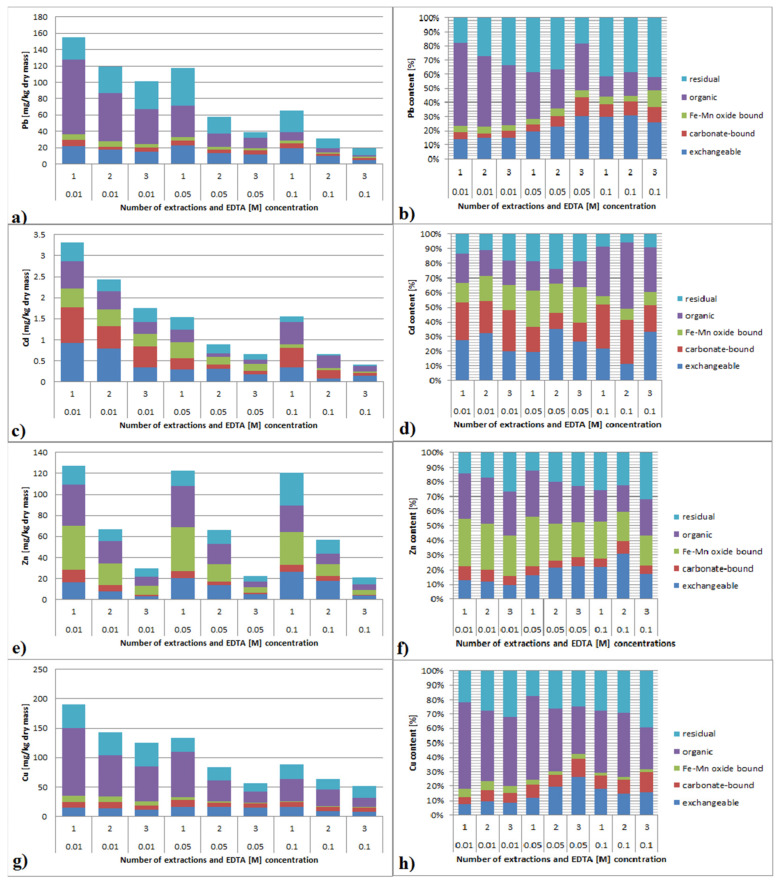
(**a**–**h**). Speciation and content of trace metals after multiple EDTA extractions.

**Table 1 ijerph-19-06978-t001:** Fractional composition of air-dried bottom sediments sample.

Grain Size [µm]	Mass [g]	Share in Total Content [%]
>2000	24.65	2.3
2000–1000	30.57	2.9
1000–500	103.8	9.8
500–250	171.16	16.2
250–90	294.88	27.9
<90	433.11	40.9

**Table 2 ijerph-19-06978-t002:** TM content, Tessier speciation, raw sediments sample.

TM	Fraction										Total
	Exchangeable		Carbonate		Fe/Mn Bound		Organic		Residual		
	mg/kg	%	mg/kg	%	mg/kg	%	mg/kg	%	mg/kg	%	mg/kg
Zn	7.2	3.0	92.0	38.9	71.5	30.1	51.8	21.8	14.8	6.2	237.3
Cd	2.9	47.0	2.2	36.2	0.3	4.6	0.2	3.7	0.5	8.5	6.1
Pb	25.1	13.3	51.0	27.0	52.5	27.8	33.0	17.4	27.4	14.5	189.0
Cu	1.0	0.3	48.8	13.8	28.5	8.1	251.4	71.3	23.1	6.5	352.8

**Table 3 ijerph-19-06978-t003:** Classification of soils for contamination with TM [45].

Classification	Total TM Content [mg/kg d. m.]			
	Pb	Zn	Cu	Cd
Non-polluted	≤100	≤300	≤150	≤4
Polluted	>100	>300	>150	>4

**Table 4 ijerph-19-06978-t004:** The content of the “labile”, “moderately labile” and “not extractable” fraction.

Zn									
EDTA [M]	0.01			0.05			0.1		
Extraction Multiplicity	1	2	3	1	2	3	1	2	3
q [mg/kg d.m.]	232.9	129.6	66.7	232.7	124	61.1	240	128.1	55.4
Q_1_ [mg/kg d.m.]	99.2	59.5	39.7	108.6	62.4	33.8	108.3	54.8	37.1
Q_2_ [mg/kg d.m.]	4.1	3.4	0.1	0.1	0.5	14.5	3.6	17.9	0.1
Q_3_ [mg/kg d.m.]	129.6	66.7	26.9	124	61.1	12.8	128.1	55.4	18.2
Cd									
EDTA [M]	0.01			0.05			0.1		
Extraction multiplicity	1	2	3	1	2	3	1	2	3
q [mg/kg d.m.]	6.25	3.23	2.33	6.25	1.6	0.44	6.71	1.79	0.66
Q_1_ [mg/kg d.m.]	0.93	0.89	0.67	3.03	1.13	0.2	3.59	1.03	0.49
Q_2_ [mg/kg d.m.]	2.09	0.01	0.22	1.62	0.03	0.02	1.33	0.1	0.03
Q_3_ [mg/kg d.m.]	3.23	2.33	1.44	1.6	0.44	0.22	1.79	0.66	0.14
Cu									
EDTA [M]	0.01			0.05			0.1		
Extraction multiplicity	1	2	3	1	2	3	1	2	3
q [mg/kg d.m.]	376.7	213.5	162.5	363.5	137.9	70.3	358.8	98.1	59
Q_1_[mg/kg d.m.]	106.7	44.5	39.7	177.1	67.5	21.1	216.6	39	17
Q_2_ [mg/kg d.m.]	56.5	6.5	0.1	48.5	0.1	0.1	44.1	0.1	0.1
Q_3_	213.5	162.5	122.7	137.9	70.3	49.1	98.1	59	41.9
Pb									
EDTA [M]	0.01			0.05			0.1		
Extraction multiplicity	1	2	3	1	2	3	1	2	3
q [mg/kg d.m.]	191.9	145	99.3	193.1	95.5	47.8	183.3	40.9	17
Q_1_ [mg/kg d.m.]	28.6	32.6	32.8	93.8	41.6	22.4	130.2	21.4	10.7
Q_2_ [mg/kg d.m.]	18.3	13.1	0.1	3.8	6.1	4.1	12.2	2.5	0.1
Q_3_ [mg/kg d.m.]	145	99.3	66.4	95.5	47.8	21.3	40.9	17	6.2

## Data Availability

Not applicable.

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
