# Peer review of "Assessment of Trace Metals Contamination, Species Distribution and Mobility in River Sediments Using EDTA Extraction"

_ijerph, 2022, doi:10.3390/ijerph19126978_

Round 1

Reviewer 1 Report

This paper feels more like a laboratory report and is lacking scientific explanations, I urge authors to add more scientific discussion to the paper.

specifically, equation 1 is missing units, some table area also missing units

Author Response

We would like to thank the Reviewer and Editor for the critical reading, as well as for helpful, relevant and constructive remarks. The manuscript was corrected and rewritten in accordance with the suggestions, which improved the quality of our paper. All changes are indicated in red. We acknowledge that these modifications definitely improve the quality of our manuscript. We hope that the changes and explanations are acceptable and satisfactory with the expectation of the Reviewer and Editor.

You can find below the details of the modifications and explanations. Thank you very much for revising our manuscript again!

Comment 1: This paper feels more like a laboratory report and is lacking scientific explanations, I urge authors to add more scientific discussion to the paper.

Answer: We thank the Reviewer for reading our manuscript very carefully. We agree with the Reviewer that adding an in-depth discussion will significantly improve the quality of the manuscript. As a result, we have added numerous modifications to the manuscript, all marked in red in the text. We have additionally supplemented the cited literature with the latest items. The number of amendments is very large, if we gave them all here, as a response to the review, its length would be slightly less than the length of the manuscript itself. Every chapter of this manuscript has been completed and reworked, including the abstract, conclusion, and bibliography. We hope that the quality of the manuscript has improved as a result of the corrections made.

Comment 2: specifically, equation 1 is missing units, some table area also missing units.

Answer: Thank you for this remark, we fully agree with it. Missing units have been added. The text was changed as follows:

“where qi represents the quantity of desorbed metal from binding location i, per mass unit of soil (mg/kg dry mass), after time t (h), Qi represents the quantity of desorbed metal from i, per mass unit of soil (mg/kg dry mass), at equilibrium, and ki is the rate constant of the extraction reaction for compartment i (1/h).

The model of multiple first-order reactions allows determining the quantity and the extraction rate of metal cations associated to each fraction. A simplification of the presented model to a two first-order reactions model enables classification of each metal cation into three fractions, Q1, Q2 and Q3 [43]. The use of the above equation allows the characterization of the three fractions, as follows [42] (2):

q=Q1+Q2+Q3                                                                                                          (2)

q – pseudo-total concentration of metal in the sediment [mg/kg dry mass].

Q1 – ‘‘labile” fraction [mg/kg dry mass],

Q2 – ‘‘moderately labile” fraction [mg/kg dry mass],

Q3 – metal fraction which is not extractable [mg/kg dry mass].

For the purposes of the calculations, it was assumed that the first two fractions obtained during the extraction according to the Tessier method are “labile”, the last one is “not extractable”. Extraction kinetics and statistical calculations were made using MS Excel. Statistical relationships were calculated by grouping the mass removal of selected TMs into pairs for all fractions, contact times, EDTA concentrations and extraction multiplicities.”

Reviewer 2 Report

Do you have data on the surface area of the sediment sample used for testing? It seems like it would be a useful addition.

Was the possible dissolution of the sample taken into account when studying the extraction kinetics?

Author Response

We would like to thank the Reviewer and Editor for the critical reading, as well as for helpful, relevant and constructive remarks. The manuscript was corrected and rewritten in accordance with the suggestions, which improved the quality of our paper. All changes are indicated in red. We acknowledge that these modifications definitely improve the quality of our manuscript. We hope that the changes and explanations are acceptable and satisfactory with the expectation of the Reviewer and Editor.

You can find below the details of the modifications and explanations. Thank you very much for revising our manuscript again!

Comment 1: Do you have data on the surface area of the sediment sample used for testing? It seems like it would be a useful addition.

Answer: Thank you for reading our manuscript carefully. Information on both the surface structure of the sediment before it is subjected to extraction and after extraction could significantly improve the understanding of the process, its mechanism and changes taking place in the sample.

Unfortunately, the research mentioned by the Reviewer has not been performed. The time that elapsed from the collection of samples and the performance of the tests was too long that we are concerned that the properties and structure of the samples have changed, so that the current state does not reflect the initial properties of the samples, which means that the performing this research now and putting the results into this manuscript is not possible. However, considering the Reviewer's suggestion to be very valuable, we will supplement our next research and publications with the analyzes suggested by the Reviewer.

Comment 2: Was the possible dissolution of the sample taken into account when studying the extraction kinetics?

Answer: The Reviewer drew attention to an important and interesting issue that was not discussed in the manuscript. Considerations on this were added to the discussion of the results. A bottom sediment sample was taken for the tests. This means that the material in its original state and environment was in constant contact with water. It can be suspected that all the compounds soluble in water dissolved in it, and the overwhelming majority of the sample components are compounds with limited water solubility. The sample, in accordance with the methodology, was dried and then extracted with aqueous solutions of various chemical compounds. During the extraction, due to the addition of the solvent, the chemical compounds constituting the sample could dissolve. However, as they are compounds with limited solubility, it was assumed that the effect of the solvent itself would be negligible. It seems that the influence of the chemicals used during extraction will be much more important here. It should be noted that EDTA is weak acid and could cause direct chemical reactions with the components constituting the sample. This means that many compounds, products of chemical reactions, had greater solubility than the parent compounds. On the one hand, the occurring chemical reactions will cause the release of metal ions into the solution, on the other hand, however, they will cause the structure of the sample to remodel, which will result in a change in the speciation image. Some of the chemical reactions taking place are quite slow which means that the assumed extraction times are too short to complete the reaction completely. This hypothesis is confirmed by the effects of the second and third extraction. After each subsequent extraction, the speciation pattern continues to change, and the mass of the sample remaining after the process should decrease. Nevertheless, this is only speculation since the sample was not weighed after the process.

Reviewer 3 Report

The manuscript describes an ineresting research, but some modifications are nedded

Please, see review pdf attached. Some general considerations are:

(1) Review and rewrite the abstract, 1st and 2nd sentences are very similar, and there are some English mistakes

(2) Review section 2.

-To clarify section 2.3, a diagram would be convenient, extraction 1, extraction 2…

-remove 2.4, and include it in 2.3

-definition of Q1, Q2 and Q3 is different in section 2.5 from the abstract and conclusion??

(3) before Table 2 define: exchangeable and bioavailable fractions

(4) put the figures 1a-f, 2a-f…4a-f, by pairs in the same line, as follow 1a,b (line 159); 1c,d; and 1e,f in the same page, to facilitate the comparison between them

(5) change 3.4 section (See review pdf attached)

(6) Include correlations graphs in the manuscript

(7) Put figures 5a,b; 5c,d…. by pairs, the same as 1a,b…

(8) review section discussion. One summary table could clarify the discussion to the reader

(9) review conclusions: regroup by methodologies, expand conclusions of correlations

Author Response

We would like to thank the Reviewer and Editor for the critical reading, as well as for helpful, relevant and constructive remarks. The manuscript was corrected and rewritten in accordance with the suggestions, which improved the quality of our paper. All changes are indicated in red. We acknowledge that these modifications definitely improve the quality of our manuscript. We hope that the changes and explanations are acceptable and satisfactory with the expectation of the Reviewer and Editor.

You can find below the details of the modifications and explanations. Thank you very much for revising our manuscript again!

General Comment 1: The manuscript describes an interesting research, but some modifications are needed.

Answer: We thank the Reviewer for reading our manuscript very carefully.

General Comment 2: Please, see review pdf attached.

Answer: Thank you for the numerous amendment proposals. We acknowledge the Reviewer's right, all suggested corrections have been applied.

Detailed comment 1: Review and rewrite the abstract, 1st and 2nd sentences are very similar, and there are some English mistakes

Answer: Abstract was rewritten, English mistakes were removed. After corrections, the abstract obtained the following wording:

The impact of the ethylenediaminetetraacetic acid (EDTA) on speciation image of selected trace metals (Zn, Cd, Cu, Pb) in bottom sediments was determined. The influence on the effectiveness of metal removal of extraction multiplicity, type of metal, extraction time, concentration of EDTA were analysed. With the increase of extraction multiplicity, concentration of EDTA and contact time the efficiency of trace metals leaching is increasing. Speciation analysis revealed that EDTA not only leached metals from bioavailable fractions but also caused the transition of the metals between the fractions. The biggest amounts of bioavailable forms were found for Cd, less for Zn. The amount of bioavailable fraction was the lowest for Cu and Pb. The two first-order kinetic models fitted well the kinetics of metals extraction with EDTA, allowing the metals fractionation into: labile (Q1), moderately labile (Q2) and not extractable fractions (Q3).

Detailed comment 2: Review section 2.

-To clarify section 2.3, a diagram would be convenient, extraction 1, extraction 2…

-remove 2.4, and include it in 2.3

-definition of Q1, Q2 and Q3 is different in section 2.5 from the abstract and conclusion??

Answer: Figure explaining Tessier speciation scheme was created. Paragraph 2.4 was removed. Text from 2.4 was moved to 2.3. The new wording of paragraph 2.3 is as follows:

Both, in the bottom sediment, prior to extraction, as well as in residual sediment after 24h extraction of HM, speciation was determined according to methodology of Tessier [33], as following: the first stage for the speciation was extraction with 1 M MgCl2 in room temperature for one hour, with pH = 7, speed of shaking 250 rpm. After filtration and filtrate collection, the remaining residue was treated for 8 hours in room temperature with 1 M CH3COONa acidified to pH = 5 with CH3COOH, speed of shaking 250 rpm. After filtrate collection, the third fraction was obtained by adding to the remaining residue 20 ml 0.04 M NH2OH * HCl in 25% CH3COOH, at the temperature of 96 +/- 3°C, for about 8 hours. The fourth fraction was obtained by adding to the remaining residue 3 ml 0.02 M HNO3 and 5 ml 8.8 M H2O2, at pH = 2, at the temperature of 85 +/- 2°C, for 2 hours. After this period the second portion of reagents was added - 3 ml 8.8 M H2O2 and heated at the temperature of 85 +/- 2°C, for 3 hours. After cooling 5 ml of 3.2 M CH3COONH4 was added in 20% HNO3, stirred for 30 minutes, speed of shaking 250 rpm. The bottom residue remaining after the filtration was mineralized with HNO3/HClO4 mixture in 5:1 ratio. As a result of the extraction, five fractions are obtained, referred to as exchangeable, carbonate, Fe/Mn bound, organic and residual. A detailed scheme of speciation is shown in Figure 1. The content of cadmium, lead, copper and zinc in every fraction was determined with Flame Atomic Absorption spectrometry – FAAS using AAnalyst 300 (Perkin Elmer, Massachusetts, USA).

Figure 1 Tessier speciation scheme /kindly check in the corrected article/

Definition of Q1, Q2 and Q3 was unified in entire text. The original nomenclature from the cited source is used.

q – pseudo-total concentration of metal in the soil [mg/kg dry mass].

Q1 – ‘‘labile” fraction [mg/kg dry mass],

Q2 – ‘‘moderately labile” fraction [mg/kg dry mass],

Q3 – metal fraction which is not extractable [mg/kg dry mass].

Detailed comment 3: before Table 2 define: exchangeable and bioavailable fractions

Answer: We agree with the Reviewer, that definitions would be useful. In the case of the exchangeable fraction, due to the previous correction suggested by the Reviewer (detailed comment 2), we added a graphic with an extraction diagram showing the nomenclature of the factions and additionally an explanatory text in chapter 2.3 was added. We believe that it would be an unnecessary repetition to give a definition of exchangeable here again. In case of bioavailability, the following text has been added to the manuscript:

Metals that are weakly bound could be available for living organisms, bioavailable. Usually, the first three fractions (exchangeable, carbonate and Fe/Mn bound) are considered to be bioavailable.

Detailed comment 4: put the figures 1a-f, 2a-f…4a-f, by pairs in the same line, as follow 1a,b (line 159); 1c,d; and 1e,f in the same page, to facilitate the comparison between them

Answer: Comment have been applied. Kindly check new layout of figures in the corrected article. In connection with the addition of a new figure (kindly check Detailed Comment 2), the numbering of figures was changed at the same time (increased by 1 in relation to the previous numbering).

Figure 2a-f. Pb extraction from bottom sediments with different EDTA concentrations (a, b: 0.01M, c, d: 0.05M, e, f: 0.1M) and extraction multiplicity

Figure 3a-f. Cd extraction from bottom sediments with different EDTA concentrations (a, b: 0.01M, c, d: 0.05M, e, f: 0.1M) and extraction multiplicity

Figure 4a-f. Zn extraction from bottom sediments with different EDTA concentrations (a, b: 0.01M, c, d: 0.05M, e, f: 0.1M) and extraction multiplicity         

Figure 5a-f. Cu extraction from bottom sediments with different EDTA concentrations (a, b: 0.01M, c, d: 0.05M, e, f: 0.1M) and extraction multiplicity   

Figure 6a-h. Speciation and content of heavy metals after multiple EDTA extraction.

Detailed comment 5: change 3.4 section (See review pdf attached)

Answer: Comment have been applied. The text has been rearranged as suggested by the Reviewer.

Detailed comment 6: Include correlations graphs in the manuscript

Answer: Comment have been applied. The figure was created as suggested by the Reviewer.

Figure 6. Correlations between trace metals extraction

Detailed comment 7: Put figures 5a,b; 5c,d…. by pairs, the same as 1a,b…

Answer: Comment have been applied, as for Detailed Comment 4.

Detailed comment 8: review section discussion. One summary table could clarify the discussion to the reader

Answer: Comment have been applied. About 50% of the new text has been added and the existing text has been corrected. The new text is as follows.

A bottom sediment sample was taken for the tests. This means that the material in its original state and environment was in constant contact with water. It can be suspected that all the compounds soluble in water dissolved in it, and the overwhelming majority of the sample components are compounds with limited water solubility. The sample, in accordance with the methodology, was dried and then extracted with aqueous solutions of various chemical compounds. During the extraction, due to the addition of the solvent, the chemical compounds constituting the sample could dissolve. However, as they are compounds with limited solubility, it was assumed that the effect of the solvent itself would be negligible. It seems that the influence of the chemicals used during extraction will be much more important here. It should be noted that EDTA is weak acid and could cause direct chemical reactions with the components constituting the sample. This means that many compounds, products of chemical reactions, had greater solubility than the parent compounds. On the one hand, the occurring chemical reactions will cause the release of metal ions into the solution, on the other hand, however, they will cause the structure of the sample to remodel, which will result in a change in the speciation image. Some of the chemical reactions taking place are quite slow which means that the assumed extraction times are too short to complete the reaction completely. This hypothesis is confirmed by the effects of the second and third extraction. After each subsequent extraction, the speciation pattern continues to change, and the mass of the sample remaining after the process should decrease. Nevertheless, this is only speculation since the sample was not weighed after the process.

EDTA is a very strong metal chelating agent. Increasing its concentration, contact time and the extraction multiplication factor significantly contributes to increasing the efficiency of TM removal from bottom sediments. This relationship is observed for all tested metals. The effectiveness of the extraction process depended on the properties of the metal and its speciation in the sediment.

EDTA extraction allows to remove not only the fractions recognized as bioavailable (exchangable, carbonate, Fe/Mn bound), but also biologically not available forms (organic and residual). The research revealed a different influence of the above-mentioned factors on the effects of the extraction process for the four metals tested. The concentration of EDTA had the strongest effect on lead leaching. Only in the case of the lowest concentration, the extension of the extraction time and the multiplicity of the extraction had an impact on the efficiency of Pb elution. Cd extraction increased with time for each EDTA concentration but was the strongest with 0.1M EDTA solution. The Cd extraction fold was also the most significant for the 0.1M EDTA solution. The Zn extraction was the same for the three EDTA concentrations and did not change significantly with time. As for Cd, the extraction of Zn from the sludge was most influenced by the extraction multiplication factor. Cu extraction increased with increasing EDTA concentration and during extraction. The copper leaching rate was the same for all three EDTA solutions. The obtained results suggest that EDTA may be an appropriate factor to determine the content of bioavailable TM in sediments. The ability to extract Zn and Cd by EDTA compared to Pb and Cu may be affected by anoxia, which results from the greater solubility of ZnS and CdS compared to CuS and PbS [47].

The use of EDTA also changes the image of the speciation of the remaining sediment. The expected effect was the leaching of metals from weakly bound fractions. However, a shift of metals from the stronger metal-binding fraction to the weaker metal-binding fraction was also observed. This phenomenon should be associated with a change in the composition and content of substances contained in the bottom sediment remaining after extraction. EDTA has the ability to remove not only metals but also other substances of sediments. The change in the matrix composition, therefore, translated into a change in the speciation image of metals. As a result, it is advisable to use another EDTA extraction cycle. In the next extraction cycle, metals in weakly bound fractions are removed and the remaining sediments are further transformed, which results in the shifting of TM from the fractions that bind the metals more strongly to the fractions that bind them weaker. However, it should be remembered that each subsequent extraction cycle rinses not only TM from the sediments, but also other substances, including carbonates, phosphates, organic compounds, etc. [48]. After extraction, some TM remain in the sediment, they are still bioavailable. The highest amounts of bioavailable forms were found for Cd, less for Zn. The amount of the bioavailable fraction was the lowest for Cu and Pb. Although the use of EDTA reduces the TM content in the treated sludge, due to a change in the matrix composition, it may lead to an increase in the toxicity and bioavailability of metals remaining afterwards [48]. Although EDTA washing could effectively remove TM, it also may result in a significant decline in sediment quality [49]. The obtained results are consistent with the ones obtained by other researchers. Cheng et al. [50], by using a blend of 0.05M EDTA with weak organic acids (citric, oxalic and tartaric), were able to remove more than 80% TM. Simultaneously using the BCR extraction scheme, Cheng et al. [50] confirm change of speciation image after EDTA extraction. The removal of TM mainly from the labile fractions was observed but also a significant reduction in the content of bound metals in the last, fourth, residual fraction. They suggested that acidification of the environment increases the efficiency of TM extraction. Ferrans et al. [51] came to similar conclusions using EDTA to remove metals from dredged sediments. Using 0.05M EDTA they removed more metals than they did with 0.01M. Lumia et al. [52] reported up to 85% removal of TM with the use of EDTA. The extraction efficiency was metal dependent, the highest for Zn. The effectiveness of 0.1M EDTA was greater than using the 0.05M solution. However, increasing the EDTA concentration to 1M did not increase the leaching efficiency, as opposed to increasing the contact time from 3 to 24 hours. The discussed results show the influence of bottom sediment components such as minerals, organic substances on the adsorption, immobilization and availability of TMs, which was confirmed by other researchers [53-56]. The extraction processes remove organic substances present in the sediments. Under these conditions, non-eluted TM bind in the crystal structures of the bottom sediment [8]. Metals bound in the mineral structures are also mobile and may migrate to other components of the environment [57].

Detailed comment 9: review conclusions: regroup by methodologies, expand conclusions of correlations

Answer: Comment have been applied. The new text is as follows.

The originality and novelty of this manuscript lies in the comparison of three approaches that are generally rarely used simultaneously in research and described separately in the literature. The metals were characterized by abundant extraction in more concentrated EDTA solutions and in the first stages of extraction. This indicates a high constant of the desorption rate and a high extraction by EDTA.

The method of sequential extraction of metals from environmental samples provides important information about the possible mobility and toxicity of metals to environmental components. The speciation analysis performed in the remains after EDTA extraction showed the migration of TM between the sediment fractions. This was especially observed in the case of copper, which has a very strong affinity for the organic fraction, and the tests performed in the remaining extraction sediments increase its share in the residual fraction. Such behavior indicates a strong dependence of the share of metals in individual fractions on the mineral-organic-microbiological composition of bottom sediments.

The extraction multiplicity and EDTA concentration as well as the contact time affect the results of TM leaching from the bottom sediments. For the same contact time and the same EDTA concentration, the highest leaching efficiency was obtained for Cd, lower for Pb and Cu, and the lowest for Zn.

The kinetic approach revealed the presence of two distinct pools in the metal-EDTA fractions and a higher desorption rate for Cd and Zn compared to Pb and Cu. Two first order kinetic models fit well with EDTA metal extraction kinetics, allowing metals to be fractionated into: “labile” (Q1), “moderately labile” (Q2) and “not extractable” fractions (Q3). The correlation study showed some significant metal synergy which was quite varied in the sediment fractions.

Three methods used in this work: the sequential extraction scheme according to Tessier, a EDTA extraction and kinetic calculations showed clearly differentiation in the availability of four tested TM occurring under the same conditions in the tested sludge.

Reviewer 4 Report

The study concerns the extraction of trace metals Cu, Zn, Cd and Pb from bottom sediments from the Utrata river with EDTA. The authors use different methods and techniques for estimating the contamination with trace metals under study, their distribution between the different compartments of the sediments as well as their mobility. The influence of EDTA extraction on the speciation of residual metals is only one parts of the study, as it is presented in the article. In this sense, I believe that the title does not accurately reflect the nature of the study. A more general title would sound much better, for example „Assessment of trace metals contamination, species distribution and mobility in river sediments using EDTA extraction”. 

In my opinion, major revision is needed on the way the results of the research are presented and discussed.

  1. The term “Heavy metals” should be avoided (Read article: Heavy metals a meaningless term (IUPAC Technical Report)). “metalloids and trace metals” is more corrected phrase.
  2. Only one sentence is included in the introduction concerning the available in the literature articles related to EDTA extraction. More and newer information needs to be provided to motivate the research.
  3. The set goal is very limited. It is not clear why some methods are used - Extraction Kinetics, Extraction Correlations etc.
  4. In “Materials and Methods” there is no information about the number of points from which the sediments were taken, as well as their quantity. It is not clear how representative the study is.
  5. There is also no information about the method for determination of correlation between the extraction of metals from bottom sediments for different metal-to-metal pairs and the EDTA concentration.
  6. Line 111” pseudo-total concentration of metal in the soil” or “the pseudo-total concentration of metal in the sediment”
  7. The "Results" section is too extended, at the expense of "Discussion"
  • In section 3.3 only figures with the percentage distribution of the element are enough by writing the total extracted concentration (in mg/kg) of the element above each column.
  • What does “mineralization” (marked on the figure 1-4) mean.
  • In the figure captions (Figs 1-4) is written “……….multiplicities (single, double and triple)”, while the figure legend contains extraction 1, extraction 2, extraction 3. Unify the terms in the text as well.
  • Try to shorten the texts in sections 3.3 and 3.6 by summarizing the similar behavior of the elements and highlighting the differences.
  • Put the units in Table 4
  • Lines from 274 to 280 repeat lines from 106 to 114
  • Figure 5 could be reduced analogous to figs 1-4
  1. The “Conclusions” must be rewritten. Clearly and concretely indicate the objectives achieved in the given experiment without general expressions. Тhe first paragraph is more appropriate for the discussion.

Author Response

We would like to thank the Reviewer and Editor for the critical reading, as well as for helpful, relevant and constructive remarks. The manuscript was corrected and rewritten in accordance with the suggestions, which improved the quality of our paper. All changes are indicated in red. We acknowledge that these modifications definitely improve the quality of our manuscript. We hope that the changes and explanations are acceptable and satisfactory with the expectation of the Reviewer and Editor.

You can find below the details of the modifications and explanations. Thank you very much for revising our manuscript again!

General Comment 1: The study concerns the extraction of trace metals Cu, Zn, Cd and Pb from bottom sediments from the Utrata river with EDTA. The authors use different methods and techniques for estimating the contamination with trace metals under study, their distribution between the different compartments of the sediments as well as their mobility. The influence of EDTA extraction on the speciation of residual metals is only one parts of the study, as it is presented in the article. In this sense, I believe that the title does not accurately reflect the nature of the study. A more general title would sound much better, for example „Assessment of trace metals contamination, species distribution and mobility in river sediments using EDTA extraction”. 

Answer: Title was changed, as suggested by the Reviewer.

General Comment 2: In my opinion, major revision is needed on the way the results of the research are presented and discussed.

Answer: Thank you for reading our manuscript carefully. We tried to improve the article as suggested by the Reviewer.

Detailed comment 1: The term “Heavy metals” should be avoided (Read article: Heavy metals a meaningless term (IUPAC Technical Report)). “metalloids and trace metals” is more corrected phrase.

Answer: We agree with the Reviewer. Term heavy metals is misleading. Comment have been applied. The term heavy metals (HM) was removed. Instead, term trace metals (TM) was used in abstract, as key word and in main body of manuscript. However, as many researchers continue to use the term heavy metals, this is still one of the keywords.

Detailed comment 2: Only one sentence is included in the introduction concerning the available in the literature articles related to EDTA extraction. More and newer information needs to be provided to motivate the research.

Answer: Comment have been applied. Introduction was expanded and more information was provided.

The natural content of trace metals (TM) in sediments is the result of geochemical construction of the basin. Currently, the observed TM concentrations are typically much higher than the natural ones and are an effect of human activity, mainly the inflow of wastewater contaminated with TM and other industrial or agricultural activities. Industrial, agricultural and mining activities are important anthropogenic sources of TM pollution [1-3]. Firstly, deposited in soil, through surface runoff and groundwater, TM reach rivers and lakes, where they are eventually deposited in bottom sediments. Accumulated in bottom sediments, metals pose a threat of secondary pollution of the reservoir. They do not undergo biological decomposition and are considered as persistent environmental pollution [4]. The migration and accumulation of metals and metalloids in the environmental components (sediment, soil) is largely dependent on pH, redox potential (Eh), types of minerals, organic carbon content and microorganisms, as well as plant species, root surface, exudate rate from roots and transpiration rate [5-7]. The immobilization and mobility of TMs in the soil environment or in bottom sediments strongly depends on their interaction with solid components, especially minerals, organic matter and microorganisms, which are the main components of solid phase aggregates [8-9]. Metal complexation with ligands and adsorption to colloidal particles strongly influence their reactivity, mobility, bioavailability and toxicity [10]. During determination of the metal contamination degree it is important to assess not only the total content of TM but additionally forms of their occurrence [11-12]. Metal ions in sediments are partitioned between the different phases, i.e., organic matter, oxyhydroxides of iron, aluminum and manganese, phyllosilicate minerals, carbonates and sulfides [13].

Speciation analysis is performed in order to determine the forms and types of compounds with which TMs are associated. [14]. As a result of speciation analysis, the information on how much metal accumulated in the sediments is available for living organisms can be obtained [15]. There are many methodologies for speciation because the methodology is designed according to the purpose to which is aimed [16]. The two most common sequential extraction methods in use are the Tessier method and the BCR method [17-23]. In the case of high concentration of metals in the sediment, it is advisable to carry out reclamation works [24-25]. There are many methods of reclamation, one of which is the leaching of metals from sediments with a chelating agent [26-28], such as ethylene diamine tetraacetic acid (EDTA) [29-38]. The use of a mixture of chelating agents can improve the efficiency of the extraction [39].

The aim of this study is to determine the changes in the speciation image as an effect of EDTA concentration, extraction time and multiplicity for selected TM. The originality and novelty of this manuscript lies in the comparison of three approaches that are generally rarely used simultaneously in research and described separately in the literature.

Detailed comment 3: The set goal is very limited. It is not clear why some methods are used - Extraction Kinetics, Extraction Correlations etc.

Answer: We add some additional descriptions to clarify extraction kinetics and correlations. our goal is also mentioned and explained in previous answer.

For the purposes of the calculations, it was assumed that the first two fractions obtained during the extraction according to the Tessier method are “labile”, the last one is “not extractable”. Extraction kinetics and statistical calculations were made using MS Excel. Statistical relationships were calculated by grouping the mass removal of selected TMs into pairs for all fractions, contact times, EDTA concentrations and extraction multiplicities.

Detailed comment 4: In “Materials and Methods” there is no information about the number of points from which the sediments were taken, as well as their quantity. It is not clear how representative the study is.

Answer: The sample was only taken at one point. Our aim was not to investigate the local state of the environment (such pollution monitoring is rather of local importance). The Utrata river was chosen as a model example of a river with heavily contaminated bottom sediments. The source of pollution (identified and thoroughly described in many previous publications) was the discharge of industrial wastewater, including heavy machinery industry, and the inflow of leachate from an uninsulated municipal landfill. The aim of our research was to determine whether EDTA, as a TM extracting factor, would be effective (according to the obtained results, it can be clearly confirmed, but on the basis of previous studies it was expected). The most important novelty of our article, which also seems to be universal and not limited to our research object (Utrata river), is the statement that the TM extraction with the use of EDTA affects the image of TM speciation in the remaining sediment. This means that EDTA not only permanently complexes TM, enabling their effective removal from bottom sediments, but also causes the transfer of TM between fractions, including in particular transferring them to less and less bound fractions, on the one hand increasing the availability of residues and, on the other hand, facilitating their further removal in the next extraction step.

Because of that, we added originality and novelty statement, at the end of introduction:

The originality and novelty of this manuscript lies in the comparison of three approaches that are generally rarely used simultaneously in research and described separately in the literature.

Detailed comment 5: There is also no information about the method for determination of correlation between the extraction of metals from bottom sediments for different metal-to-metal pairs and the EDTA concentration.

Answer: We believe this comment is very similar to Detailed comment 3. We think the following amendment will clarify the method we used. For the purposes of the calculations, it was assumed that the first two fractions obtained during the extraction according to the Tessier method are “labile”, the last one is “not extractable”. Extraction kinetics and statistical calculations were made using MS Excel. Statistical relationships were calculated by grouping the mass removal of selected TMs into pairs for all fractions, contact times, EDTA concentrations and extraction multiplicities.

Detailed comment 6: Line 111” pseudo-total concentration of metal in the soil” or “the pseudo-total concentration of metal in the sediment”

Answer: Comment have been applied. Of course we are referring to the sediments, word sediment is used instead of soil.

Detailed comment 7: The "Results" section is too extended, at the expense of "Discussion"

  • In section 3.3 only figures with the percentage distribution of the element are enough by writing the total extracted concentration (in mg/kg) of the element above each column.
  • What does “mineralization” (marked on the figure 1-4) mean.
  • In the figure captions (Figs 1-4) is written “……….multiplicities (single, double and triple)”, while the figure legend contains extraction 1, extraction 2, extraction 3. Unify the terms in the text as well.
  • Try to shorten the texts in sections 3.3 and 3.6 by summarizing the similar behavior of the elements and highlighting the differences.
  • Put the units in Table 4
  • Lines from 274 to 280 repeat lines from 106 to 114
  • Figure 5 could be reduced analogous to figs 1-4

Answer: Thank you for detailed comments.

Ad 1) We agree with the Reviewer that it is possible to report metal removal efficiency using only one figure, not two (percentage removal vs mass removal). Giving the percentage distribution and the initial content with the knowledge of the basics of mathematics allows for any calculations. However, it seems to us that for the convenience of the reader and a faster overview of the effectiveness of the process being carried out, it is worth showing both methods. Therefore, we ask the Reviewer for consent to leave both methods of data presentation on Figures.

Ad 2, 3) We agree with the Reviewer that it may not be clear enough. We have added the following text for clarification.

The amount of TM removed in a single extraction is described as extraction 1, the amount of TM removed in a double extraction is described as extraction 2, the amount of TM removed in a triple extraction is described as extraction 3. The TM content remaining after the triple extraction was marked as mineralization.

The nomenclature (single, double, triple vs extraction 1/2/3) used was standardized. Extraction 1, 2 and 3 are used both in the figures and in the text.

Extraction fold (single, double, triple) designations have been removed from the figure description. New figure captions (figures 2-5) are as follows:

Figure Xa-f. TM extraction from bottom sediments with different EDTA concentrations (a, b: 0.01M, c, d: 0.05M, e, f: 0.1M) and extraction multiplicity

Ad 4) We agree with the Reviewer that the description in 3.3 and 3.6 is quite long. The research we performed had a large scope, in the article we tried to accurately present the obtained results.

In these chapters, we tried to describe the behavior of each metal depending on the complex conditions of the experiment. We have made an attempt to rearrange the text. However, it turned out and what is confirmed by statistical analyzes from 3.5, that one general and simple universal conclusion cannot be drawn for all metals. If it were possible to draw conclusions about the same behavior of two or more metals, we would obtain very clear statistical relationships. In some cases, even significant relationships can be seen (Cd-Cu, single extraction), but by minimally changing the process conditions (increasing the concentration or the multiplication factor of the extraction), the relationship decreases significantly. Accepting a rightful suggestion of the Reviewer, we changed and expanded the discussion in which we tried to describe the similarities and differences between individual TMs, while referring to the literature.

Ad 5) The comment has been applied. We added units to table 4.

Ad 6) The comment has been applied. The unnecessary repetition has been removed.

Ad 7) As in the case of the Reviewer's previous suggestion (bullet point no 1), we agree with the Reviewer that it is possible to report speciation using only one figure, not two (percentage vs mass). Giving the percentage distribution and the initial content with the knowledge of the basics of mathematics allows for any calculations. However, it seems to us that for the convenience of the reader and a faster overview of the effectiveness of the process being carried out, it is worth showing both methods. Therefore, we ask the Reviewer for consent to leave both methods of data presentation on Figures. However, in order to save space and at the same time facilitate the comparison, we rearranged the layout, placing the individual parts of the figure not one by one, but in pairs next to each other, so that on the left there was a system describing the mass, and on the right - a percentage distribution.

Detailed comment 8: The “Conclusions” must be rewritten. Clearly and concretely indicate the objectives achieved in the given experiment without general expressions. Тhe first paragraph is more appropriate for the discussion.

Answer: We agree with the Reviewer. Conclusions paragraph was rewritten.

The originality and novelty of this manuscript lies in the comparison of three approaches that are generally rarely used simultaneously in research and described separately in the literature. The metals were characterized by abundant extraction in more concentrated EDTA solutions and in the first stages of extraction. This indicates a high constant of the desorption rate and a high extraction by EDTA.

The method of sequential extraction of metals from environmental samples provides important information about the possible mobility and toxicity of metals to environmental components. The speciation analysis performed in the remains after EDTA extraction showed the migration of TM between the sediment fractions. This was especially observed in the case of copper, which has a very strong affinity for the organic fraction, and the tests performed in the remaining extraction sediments increase its share in the residual fraction. Such behavior indicates a strong dependence of the share of metals in individual fractions on the mineral-organic-microbiological composition of bottom sediments.

The extraction multiplicity and EDTA concentration as well as the contact time affect the results of TM leaching from the bottom sediments. For the same contact time and the same EDTA concentration, the highest leaching efficiency was obtained for Cd, lower for Pb and Cu, and the lowest for Zn.

The kinetic approach revealed the presence of two distinct pools in the metal-EDTA fractions and a higher desorption rate for Cd and Zn compared to Pb and Cu. Two first order kinetic models fit well with EDTA metal extraction kinetics, allowing metals to be fractionated into: “labile” (Q1), “moderately labile” (Q2) and “not extractable” fractions (Q3). The correlation study showed some significant metal synergy which was quite varied in the sediment fractions.

Three methods used in this work: the sequential extraction scheme according to Tessier, a EDTA extraction and kinetic calculations showed clearly differentiation in the availability of four tested TM occurring under the same conditions in the tested sludge.

Reviewer 5 Report

Based on information of the manuscript titled “EDTA Extraction Influence on Residual Heavy Metals Speciation Forms in Bottom Sediments” it is the recommendation of this reviewer that major changes are required to be addressed.

  • Keywords should have key word that not in the title so that the target audiences have variety words to search for the manuscript.
  • expand the introduction with relevant information
  • Novelty of the paper should be highlighted.
  • Methods do not describe the statistical analysis. Please add it as well.
  • The results and discussion are extremely poor, more research in similar studies is needed. Authors can compare their results with literature’s results.
  • References could be updated with recent years publications
  • The manuscript should be proofread, there are several typos and grammatical mistakes.
  • Improve the quality of the figures (see Fig 1.a) the "a" is not legible.
  • Edit Fig.5, you are using “,” to separate decimals
  • Line 62. The writing is not clear.
  • Line 38: “Because of that, speciation analysis is performed” improve the way to start the paragraph.
  • Line 263-273. Integrate both paragraphs since it refers to a comparison with the same author.

Author Response

We would like to thank the Reviewer and Editor for the critical reading, as well as for helpful, relevant and constructive remarks. The manuscript was corrected and rewritten in accordance with the suggestions, which improved the quality of our paper. All changes are indicated in red. We acknowledge that these modifications definitely improve the quality of our manuscript. We hope that the changes and explanations are acceptable and satisfactory with the expectation of the Reviewer and Editor.

You can find below the details of the modifications and explanations. Thank you very much for revising our manuscript again!

General Comment: Based on information of the manuscript titled “EDTA Extraction Influence on Residual Heavy Metals Speciation Forms in Bottom Sediments” it is the recommendation of this reviewer that major changes are required to be addressed.

Answer: Thank you for reading our manuscript carefully. We tried to include all comments from the Reviewer and correct this manuscript as best as possible.

Detailed comment 1: Keywords should have key word that not in the title so that the target audiences have variety words to search for the manuscript.

Answer: Title as well as key words were changed. New title is: Assessment of trace metals contamination, species distribution and mobility in river sediments using EDTA extraction. New key words are: EDTA leaching; bottom sediments; trace metals; heavy metals; speciation; residual trace metals.

Detailed comment 2: expand the introduction with relevant information

Answer: Introduction was rewritten.

The natural content of trace metals (TM) in sediments is the result of geochemical construction of the basin. Currently, the observed TM concentrations are typically much higher than the natural ones and are an effect of human activity, mainly the inflow of wastewater contaminated with TM and other industrial or agricultural activities. Industrial, agricultural and mining activities are important anthropogenic sources of TM pollution [1-3]. Firstly, deposited in soil, through surface runoff and groundwater, TM reach rivers and lakes, where they are eventually deposited in bottom sediments. Accumulated in bottom sediments, metals pose a threat of secondary pollution of the reservoir. They do not undergo biological decomposition and are considered as persistent environmental pollution [4]. The migration and accumulation of metals and metalloids in the environmental components (sediment, soil) is largely dependent on pH, redox potential (Eh), types of minerals, organic carbon content and microorganisms, as well as plant species, root surface, exudate rate from roots and transpiration rate [5-7]. The immobilization and mobility of TMs in the soil environment or in bottom sediments strongly depends on their interaction with solid components, especially minerals, organic matter and microorganisms, which are the main components of solid phase aggregates [8-9]. Metal complexation with ligands and adsorption to colloidal particles strongly influence their reactivity, mobility, bioavailability and toxicity [10]. During determination of the metal contamination degree it is important to assess not only the total content of TM but additionally forms of their occurrence [11-12]. Metal ions in sediments are partitioned between the different phases, i.e., organic matter, oxyhydroxides of iron, aluminum and manganese, phyllosilicate minerals, carbonates and sulfides [13].

Speciation analysis is performed in order to determine the forms and types of compounds with which TMs are associated. [14]. As a result of speciation analysis, the information on how much metal accumulated in the sediments is available for living organisms can be obtained [15]. There are many methodologies for speciation because the methodology is designed according to the purpose to which is aimed [16]. The two most common sequential extraction methods in use are the Tessier method and the BCR method [17-23]. In the case of high concentration of metals in the sediment, it is advisable to carry out reclamation works [24-25]. There are many methods of reclamation, one of which is the leaching of metals from sediments with a chelating agent [26-28], such as ethylene diamine tetraacetic acid (EDTA) [29-38]. The use of a mixture of chelating agents can improve the efficiency of the extraction [39].

The aim of this study is to determine the changes in the speciation image as an effect of EDTA concentration, extraction time and multiplicity for selected TM. The originality and novelty of this manuscript lies in the comparison of three approaches that are generally rarely used simultaneously in research and described separately in the literature.

Detailed comment 3: Novelty of the paper should be highlighted.

Answer: Thank you for pointing this out. As the last part of the introduction, a sentence describing the scientific novelty of the article has been added.

The originality and novelty of this manuscript lies in the comparison of three approaches that are generally rarely used simultaneously in research and described separately in the literature.

Detailed comment 4: Methods do not describe the statistical analysis. Please add it as well.

Answer: Extraction kinetics and statistical calculations were made using MS Excel. Relevant information has been added to section 2.4

For the purposes of the calculations, it was assumed that the first two fractions obtained during the extraction according to the Tessier method are “labile”, the last one is “not extractable”. Extraction kinetics and statistical calculations were made using MS Excel. Statistical relationships were calculated by grouping the mass removal of selected TMs into pairs for all fractions, contact times, EDTA concentrations and extraction multiplicities.

Detailed comment 5: The results and discussion are extremely poor, more research in similar studies is needed. Authors can compare their results with literature’s results.

Answer: Results and discussion chapters were rewritten.

The greatest emphasis on corrections was placed on the description of the extraction kinetics.

Previous studies have shown [42-43], that 24h extractions are sufficient to allow reaching an equilibrium state in sediments [31]. Santos et al. [43] found that according to Labanowski et al. [15] the readily ‘‘labile” pool using EDTA overestimates the leachability of metals in soils, particularly in the case of Pb. They observed that Pb was the least mobile metal in soil, but it was the one with the highest ‘‘labile” pool determined by EDTA extraction. According to the literature, EDTA releases Pb from several soil compartments, particularly Pb associated to Fe and Mn oxides and to organic matter. Santos et al. [43] found that there is a lack of studies about the relationship between kinetically labile fractions towards EDTA extraction and fractions associated to different soil compartments, determined by sequential extraction. However, Gismera et al. [46] obtained a good correlation between the Zn labile fraction (Q1) associated to EDTA and the exchangeable and carbonate bound fraction determined by the BCR sequential extraction scheme.

In order to simulate the kinetics of metal desorption, Fangueiro et al., [42] proposed model describing 3 fractions: ‘‘labile”, ‘‘moderately labile” and „not extractable”. The content of ‘‘labile”, ‘‘moderately labile” and „not extractable” fractions, based on kinetic models for different TM end extraction multiplicity is shown in Table 4.

A general two-step desorption process was observed in the variability of the metal cation desorption rate in all samples, indicating two kinetically distinguishable pools which correspond to two metal fractions characterized by two different desorption rates, i.e. a high desorption rate at the start of leaching followed by a reduction in reaction time.

The “labile” (Q1) and “moderately labile” (Q2) metal fractions were estimated by kinetic simulation for all samples (Table 4). The content of the labile and less labile fractions is expressed as fractions of the sum of the sediment content for comparison with the metal fractions obtained by single speciation for different EDTA concentrations and three-fold.

The discussion has been completely rebuilt.

A bottom sediment sample was taken for the tests. This means that the material in its original state and environment was in constant contact with water. It can be suspected that all the compounds soluble in water dissolved in it, and the overwhelming majority of the sample components are compounds with limited water solubility. The sample, in accordance with the methodology, was dried and then extracted with aqueous solutions of various chemical compounds. During the extraction, due to the addition of the solvent, the chemical compounds constituting the sample could dissolve. However, as they are compounds with limited solubility, it was assumed that the effect of the solvent itself would be negligible. It seems that the influence of the chemicals used during extraction will be much more important here. It should be noted that EDTA is weak acid and could cause direct chemical reactions with the components constituting the sample. This means that many compounds, products of chemical reactions, had greater solubility than the parent compounds. On the one hand, the occurring chemical reactions will cause the release of metal ions into the solution, on the other hand, however, they will cause the structure of the sample to remodel, which will result in a change in the speciation image. Some of the chemical reactions taking place are quite slow which means that the assumed extraction times are too short to complete the reaction completely. This hypothesis is confirmed by the effects of the second and third extraction. After each subsequent extraction, the speciation pattern continues to change, and the mass of the sample remaining after the process should decrease. Nevertheless, this is only speculation since the sample was not weighed after the process.

EDTA is a very strong metal chelating agent. Increasing its concentration, contact time and the extraction multiplication factor significantly contributes to increasing the efficiency of TM removal from bottom sediments. This relationship is observed for all tested metals. The effectiveness of the extraction process depended on the properties of the metal and its speciation in the sediment.

EDTA extraction allows to remove not only the fractions recognized as bioavailable (exchangable, carbonate, Fe/Mn bound), but also biologically not available forms (organic and residual). The research revealed a different influence of the above-mentioned factors on the effects of the extraction process for the four metals tested. The concentration of EDTA had the strongest effect on lead leaching. Only in the case of the lowest concentration, the extension of the extraction time and the multiplicity of the extraction had an impact on the efficiency of Pb elution. Cd extraction increased with time for each EDTA concentration but was the strongest with 0.1M EDTA solution. The Cd extraction fold was also the most significant for the 0.1M EDTA solution. The Zn extraction was the same for the three EDTA concentrations and did not change significantly with time. As for Cd, the extraction of Zn from the sludge was most influenced by the extraction multiplication factor. Cu extraction increased with increasing EDTA concentration and during extraction. The copper leaching rate was the same for all three EDTA solutions. The obtained results suggest that EDTA may be an appropriate factor to determine the content of bioavailable TM in sediments. The ability to extract Zn and Cd by EDTA compared to Pb and Cu may be affected by anoxia, which results from the greater solubility of ZnS and CdS compared to CuS and PbS [47].

The use of EDTA also changes the image of the speciation of the remaining sediment. The expected effect was the leaching of metals from weakly bound fractions. However, a shift of metals from the stronger metal-binding fraction to the weaker metal-binding fraction was also observed. This phenomenon should be associated with a change in the composition and content of substances contained in the bottom sediment remaining after extraction. EDTA has the ability to remove not only metals but also other substances of sediments. The change in the matrix composition, therefore, translated into a change in the speciation image of metals. As a result, it is advisable to use another EDTA extraction cycle. In the next extraction cycle, metals in weakly bound fractions are removed and the remaining sediments are further transformed, which results in the shifting of TM from the fractions that bind the metals more strongly to the fractions that bind them weaker. However, it should be remembered that each subsequent extraction cycle rinses not only TM from the sediments, but also other substances, including carbonates, phosphates, organic compounds, etc. [48]. After extraction, some TM remain in the sediment, they are still bioavailable. The highest amounts of bioavailable forms were found for Cd, less for Zn. The amount of the bioavailable fraction was the lowest for Cu and Pb. Although the use of EDTA reduces the TM content in the treated sludge, due to a change in the matrix composition, it may lead to an increase in the toxicity and bioavailability of metals remaining afterwards [48]. Although EDTA washing could effectively remove TM, it also may result in a significant decline in sediment quality [49]. The obtained results are consistent with the ones obtained by other researchers. Cheng et al. [50], by using a blend of 0.05M EDTA with weak organic acids (citric, oxalic and tartaric), were able to remove more than 80% TM. Simultaneously using the BCR extraction scheme, Cheng et al. [50] confirm change of speciation image after EDTA extraction. The removal of TM mainly from the labile fractions was observed but also a significant reduction in the content of bound metals in the last, fourth, residual fraction. They suggested that acidification of the environment increases the efficiency of TM extraction. Ferrans et al. [51] came to similar conclusions using EDTA to remove metals from dredged sediments. Using 0.05M EDTA they removed more metals than they did with 0.01M. Lumia et al. [52] reported up to 85% removal of TM with the use of EDTA. The extraction efficiency was metal dependent, the highest for Zn. The effectiveness of 0.1M EDTA was greater than using the 0.05M solution. However, increasing the EDTA concentration to 1M did not increase the leaching efficiency, as opposed to increasing the contact time from 3 to 24 hours. The discussed results show the influence of bottom sediment components such as minerals, organic substances on the adsorption, immobilization and availability of TMs, which was confirmed by other researchers [53-56]. The extraction processes remove organic substances present in the sediments. Under these conditions, non-eluted TM bind in the crystal structures of the bottom sediment [8]. Metals bound in the mineral structures are also mobile and may migrate to other components of the environment [57].

Detailed comment 6: References could be updated with recent years publications

Answer: References were updated. New items were added:

  1. Yu, H.; Hou, J.; Dang, Q.; Cui, D.; Xi, B.; Tan, W. Decrease in bioavailability of soil heavy metals caused by the presence of microplastics varies across aggregate levels. J. Hazard. Mater. 2020, 395, 122690.
  2. Ma, L.; Xiao, T.; Ning, Z.; Liu, Y.; Chen, H.; Peng, J. Pollution and health risk assessment of toxic metal(loid)s in soils under different land use in sulphide mineralized areas. Sci. Total Environ. 2020, 724, 138176.
  3. Kelepertzis, E.; Argyraki, A.; Chrastny, V.; Botsou, F.; Skordas, K.; Komarek, M.; Fouskas, A. Metal(loid) and isotopic tracing of Pb in soils, road and house dusts from the industrial area of Volos (central Greece). Sci. Total Environ. 2020, 725, 138300.
  4. Galunin, E.; Ferreti, J.; Zapelini, I.; Vieira, I.; Teixeira Tarley, C.R.; Abrao, T.; Santos, M.J. Cadmium mobility in sediments and soils from a coal mining area on Tibagi River watershed: environmental risk assessment. J. Hazard. Mater. 2014, 265, 280-287.
  5. Zhu, H.; Chen, L.; Xing, W.; Ran, S.; Wei, Z.; Amee, M.; Wassie, M.; Niu, H.; Tang, D.; Sun, J.; Du, D.; Yao, J.; Hou, H.; Chen, K. Phytohormones-induced senescence efficiently promotes the transport of cadmium from roots into shoots of plants: a novel strategy for strengthening of phytoremediation. J. Hazard. Mater. 2020, 388, 122080.
  6. Xu, D.M.; Fu, R.B.; Wang, J.X.; Shi, Y.X.; Guo, X.P. Chemical stabilization remediation for heavy metals in contaminated soils on the latest decade: available stabilizing materials and associated evaluation methods-a critical review. J. Clean. Prod. 2021, 321, 128730.
  7. Uddin, M.K. A review on the adsorption of heavy metals by clay minerals, with special focus on the past decade. Chem. Eng. J. 2017, 308, 438-462.
  8. Chen, H.; Xu, J.; Tan, W.; Fang, L. Lead binding to wild metal-resistant bacteria analyzed by ITC and XAFS spectroscopy. Environ. Pollut. 2019, 250, 118-126.
  9. McCready, S.; Birch, G.F.; Taylor, S.E. Extraction of heavy metals in Sydney Harbour sediments using 1M HCl and 0.05M EDTA and implications for sediment?quality guidelines, Aust. J. Earth Sci. 2003, 50, 249-255.
  10. Bolan, N.; Kunhikrishn, S.A.; Thangarajan, R.; Kumpiene, J.; Park, J.; Makino, T.; Kirkham, M.B.; Scheckel, K. Remediation of heavy metal (loid)s contaminated soils-to mobilize or to immobilize? J. Hazard. Mater. 2014, 266, 141-166.
  11. Gräfe, M.; Donner, E.; Collins, R.N.; Lombi, E. Speciation of metal(loid)s in environmental samples by X-ray absorption spectroscopy: a critical review. Anal. Chim. Acta 2014, 822, 1-22.
  12. Chen, L.; Liu, J.R.; Hu, W.F.; Gao, J.; Yang, J.Y. Vanadium in soil-plant system: source, fate, toxicity, and bioremediation. J. Hazard. Mater. 2021, 405, 124200.
  13. Li, Z.; Liang, Y.; Hu, H.; Shaheen, S.M.; Zhong, H.; Tack, F.M.; Wu, M.; Li, Y.; Gao, Y.; Rinklebe, J.; Zhao, J. Speciation, transportation, and pathways of cadmium in soil-rice systems: a review on the environmental implications and remediation approaches for food safety. Environ. Int. 2021, 156, 106749.
  14. Falaciński P.; Wojtkowska M. The use of extraction methods to assess the immobilization of metals in hardening slurries, Arch. Environ. Prot. 2021, 47, 60-70.

According to the additions, the numbering of citations has been corrected.

Detailed comment 7: The manuscript should be proofread, there are several typos and grammatical mistakes.

Answer: The manuscript has been checked by a English native speaker with academic experience. Grammatical errors and typos have been removed.

Detailed comment 8: Improve the quality of the figures (see Fig 1.a) the "a" is not legible.

Answer: The figures have been corrected. Their arrangement has been changed, typos have been removed.

Detailed comment 9: Edit Fig.5, you are using “,” to separate decimals

Answer: Comment have been applied, figure was corrected.

Detailed comment 10: Line 62. The writing is not clear.

Answer: The sentence has been rebuilt for greater clarity.

The sediment was collected from the surface layer depth of 0 - 3 cm using Kajak sampler (KC Denmark Research Equipment, Denmark). A sample was transferred to a polyethylene container and then air-dried. The sample was then sieved through a set of sieves. Only the smallest fractions were collected for the tests.

Detailed comment 11: Line 38: “Because of that, speciation analysis is performed” improve the way to start the paragraph.

Answer: The sentence has been rebuilt for greater clarity.

Speciation analysis is performed in order to determine the forms and types of compounds with which TMs are associated. [14].

Detailed comment 12: Line 263-273. Integrate both paragraphs since it refers to a comparison with the same author.

Answer: The authors had a problem with finding the correct fragment of the text, as there was probably an error in the line numbers indicated by the Reviewer. We checked the manuscript carefully and it seems to us that chapter 3.4 was meant. As suggested by the Reviewer, this fragment has been rebuilt.

Previous studies have shown [42-43], that 24h extractions are sufficient to allow reaching an equilibrium state in sediments [31]. Santos et al. [43] found that according to Labanowski et al. [15] the readily ‘‘labile” pool using EDTA overestimates the leachability of metals in soils, particularly in the case of Pb. They observed that Pb was the least mobile metal in soil, but it was the one with the highest ‘‘labile” pool determined by EDTA extraction. According to the literature, EDTA releases Pb from several soil compartments, particularly Pb associated to Fe and Mn oxides and to organic matter. Santos et al. [43] found that there is a lack of studies about the relationship between kinetically labile fractions towards EDTA extraction and fractions associated to different soil compartments, determined by sequential extraction. However, Gismera et al. [46] obtained a good correlation between the Zn labile fraction (Q1) associated to EDTA and the exchangeable and carbonate bound fraction determined by the BCR sequential extraction scheme.

In order to simulate the kinetics of metal desorption, Fangueiro et al., [42] proposed model describing 3 fractions: ‘‘labile”, ‘‘moderately labile” and „not extractable”. The content of ‘‘labile”, ‘‘moderately labile” and „not extractable” fractions, based on kinetic models for different TM end extraction multiplicity is shown in Table 4.

A general two-step desorption process was observed in the variability of the metal cation desorption rate in all samples, indicating two kinetically distinguishable pools which correspond to two metal fractions characterized by two different desorption rates, i.e. a high desorption rate at the start of leaching followed by a reduction in reaction time.

The “labile” (Q1) and “moderately labile” (Q2) metal fractions were estimated by kinetic simulation for all samples (Table 4). The content of the labile and less labile fractions is expressed as fractions of the sum of the sediment content for comparison with the metal fractions obtained by single speciation for different EDTA concentrations and three-fold.

Round 2

Reviewer 3 Report

The article has improved a lot, but needs some small modifications (see revised pdf)

Author Response

Dear Reviewer,

We would like to thank the Reviewer for the critical reading, as well as for helpful, relevant and constructive remarks. We also thank you for considering publishing our manuscript. The manuscript was corrected and rewritten in accordance with the suggestions, which improved the quality of our paper.

In the second round of review, we started by changing the text color of our manuscript to solid black. We have included all the corrections made in the previous round in the consolidated text of the manuscript. We are introducing new corrections as suggested by the Editor using the “Track Changes” function using MS Word. We acknowledge that these modifications definitely improve the quality of our manuscript. We hope that the changes and explanations are acceptable and satisfactory with the expectation of the Reviewer.

Reviewer can find below the details of the modifications and explanations. Thank you very much for revising our manuscript again!

Comment 1: 2.4 and 3.4 have the same name, please differentiate to avoid ambiguity.

Answer: We agree with the Reviewer. 3.4 was changed to: Lability determination.

Comment 2: good figure, but I think it needs some changes to clarify:

-mark with a vertical line the nine values of each pair of metals

-put the pairs of metals uncovered with the bars of the chart.

Answer: We agree with the Reviewer, it is worth applying the indicated corrections, the figure will be clearer. The figure was changed as suggested by the Reviewer.

Reviewer 4 Report

I agree with the corrected version and accept the answers of the authors

Author Response

Dear Reviewer,

We would like to thank the Reviewer for the critical reading, as well as for helpful, relevant and constructive remarks. We also thank you for considering publishing our manuscript. The manuscript was corrected and rewritten in accordance with the suggestions, which improved the quality of our paper.

In the second round of review, we started by changing the text color of our manuscript to solid black. We have included all the corrections made in the previous round in the consolidated text of the manuscript. We are introducing new corrections as suggested by the Editor using the “Track Changes” function using MS Word. We acknowledge that these modifications definitely improve the quality of our manuscript. We hope that the changes and explanations are acceptable and satisfactory with the expectation of the Reviewer.

Reviewer 5 Report

The manuscript has improved significantly.

- Fig. 6; improve the presentation of the information of the axes

- Table 5 and Fig. 6 are redundant

- citation format should be reviewed

Author Response

Dear Reviewer,

We would like to thank the Reviewer for the critical reading, as well as for helpful, relevant and constructive remarks. We also thank you for considering publishing our manuscript. The manuscript was corrected and rewritten in accordance with the suggestions, which improved the quality of our paper.

In the second round of review, we started by changing the text color of our manuscript to solid black. We have included all the corrections made in the previous round in the consolidated text of the manuscript. We are introducing new corrections as suggested by the Editor using the “Track Changes” function using MS Word. We acknowledge that these modifications definitely improve the quality of our manuscript. We hope that the changes and explanations are acceptable and satisfactory with the expectation of the Reviewer.

Reviewer can find below the details of the modifications and explanations. Thank you very much for revising our manuscript again!

General Comment 1: The manuscript has improved significantly.

Answer: Thank you.

Detailed comment 1:  Fig. 6; improve the presentation of the information of the axes

Answer: We agree with the Reviewer, it is worth applying corrections, the figure will be clearer. The figure was changed as suggested by the Reviewer.

Detailed Comment 2: Table 5 and Fig. 6 are redundant.

Answer: We agree with the Reviewer. Due to the changes of Figure 6 and including the data so far visible in Table 5, we decided to delete the table 5 so as not to duplicate the data.

Detailed Comment 3:  citation format should be reviewed

Answer: Citation format was checked and corrected according to instructions:

https://www.mdpi.com/journal/ijerph/instructions#references.